# REX1 is the critical target of RNF12 in imprinted X chromosome inactivation in mice

Cristina Gontan[1], Hegias Mira-Bontenbal[1], Aristea Magaraki[1], Catherine Dupont[1], Tahsin Stefan Barakat[1], Eveline Rentmeester[1], Jeroen Demmers[2] & Joost Gribnau[1]

In mice, imprinted X chromosome inactivation (iXCI) of the paternal X in the pre-implantation embryo and extraembryonic tissues is followed by X reactivation in the inner cell mass (ICM) of the blastocyst to facilitate initiation of random XCI (rXCI) in all embryonic tissues. RNF12 is an E3 ubiquitin ligase that plays a key role in XCI. RNF12 targets pluripotency protein REX1 for degradation to initiate rXCI in embryonic stem cells (ESCs) and loss of the maternal copy of *Rnf12* leads to embryonic lethality due to iXCI failure. Here, we show that loss of *Rex1* rescues the rXCI phenotype observed in *Rnf12*$^{-/-}$ ESCs, and that REX1 is the prime target of RNF12 in ESCs. Genetic ablation of *Rex1* in *Rnf12*$^{-/-}$ mice rescues the *Rnf12*$^{-/-}$ iXCI phenotype, and results in viable and fertile *Rnf12*$^{-/-}$:*Rex1*$^{-/-}$ female mice displaying normal iXCI and rXCI. Our results show that REX1 is the critical target of RNF12 in XCI.

[1] Department of Developmental Biology, Oncode Institute, Erasmus MC, PO Box 2040, 3000 CA, Rotterdam, The Netherlands. [2] Center for Proteomics, Erasmus MC, PO Box 2040, 3000 CA, Rotterdam, The Netherlands. These authors contributed equally: Cristina Gontan, Hegias Mira-Bontenbal. Correspondence and requests for materials should be addressed to J.G. (email: j.gribnau@erasmusmc.nl)

Evolution of the eutherian sex chromosomes and the concomitant gradual loss of nearly all ancestral genes from the Y chromosome forced co-evolution of intricate dosage compensation mechanisms including X chromosome inactivation (XCI). XCI leads to equalization of X-linked gene dosage between male and female cells by inactivation of one X chromosome in every female somatic cell[1]. Two different types of XCI have been described in mice. Imprinted X chromosome inactivation (iXCI) takes place during pre-implantation development in the embryo and in the extraembryonic tissues, where the paternal X chromosome is always inactivated[2,3]. At the blastocyst stage, the inactivated paternal X is reactivated within the pluripotent cells of the inner cell mass (ICM)[4], while extraembryonic tissues such as the placenta and visceral yolk sac endoderm (VYSE) retain an inactive paternal X chromosome. Upon formation of the epiblast, the cells of the embryo inactivate their maternal or paternal X chromosome (Xm and Xp, respectively) through random X chromosome inactivation (rXCI). Later during development, the inactive X (Xi) chromosome is reactivated in female primordial germ cells (PGCs) to erase the inactive state prior to conception[5].

iXCI and rXCI utilize complex regulatory networks to properly induce mono-allelic *Xist* expression from one X chromosome. *Xist* is transcribed in a 17-kb-long non-coding RNA that spreads in *cis* to coat the future Xi chromosome, initiating epigenetic changes including H3K27me3 accumulation, involved in establishment and maintenance of the inactive state (reviewed in ref. [6]). *Rnf12*, located in close proximity to *Xist*, plays a crucial role in the regulation of iXCI[7,8]. Maternal transmission of an *Rnf12* mutant allele to daughters is lethal, due to failure of the Xp to inactivate during pre-implantation development. On the other hand, daughters with a paternally transmitted *Rnf12* mutant allele are viable and do not show iXCI defects[7]. How RNF12 mechanistically effects iXCI in vivo is still an open question. In addition, rXCI is severely affected upon differentiation of $Rnf12^{-/-}$ embryonic stem cells (ESCs), while *Rnf12* heterozygous ESCs manage to inactivate an X chromosome, indicating that one functional copy of *Rnf12* is required to properly initiate rXCI in vitro[9,10].

*Rnf12* encodes an E3 ubiquitin ligase, and pull-down experiments of RNF12 followed by mass spectrometry identified REX1 as a partner and target of RNF12 in ESCs[11]. The role of REX1 in pluripotency of ESCs, in genomic imprinting and in pre-implantation development has been studied in mice[12–14]. *Rex1* arose in placental mammals via retrotransposition of the constitutively expressed YY1 transcription factor[12]. In rXCI, REX1 acts by regulating *Tsix* and *Xist* expression in mouse ESCs and dose-dependent breakdown of REX1 facilitates female-exclusive initiation of rXCI in differentiating ESCs[11,15]. Whether RNF12 acts in iXCI through REX1 is unknown. Also, putative roles for *Rex1* in rXCI and X chromosome reactivation (XCR) in vivo have not been studied so far.

Here, we dissect the *Rex1-Rnf12* axis in XCI in vivo and in vitro. We show that REX1 is the prime target of RNF12 in ESCs. We also show that deletion of *Rex1* in $Rnf12^{-/-}$ ESCs rescues the XCI phenotype, indicating that, at least in vitro, RNF12 regulates rXCI primarily through REX1. Moreover, the lethal phenotype of $Rnf12^{-/+}$ (in the −/+ or +/− nomenclature, the maternally inherited allele is shown first) and $Rnf12^{-/-}$ female mice is completely rescued in a mutant *Rex1* background, indicating that RNF12-mediated degradation of REX1 is also a critical event in iXCI. These results highlight the crucial role for RNF12 in facilitating initiation of rXCI and iXCI, by targeting REX1 for proteasomal degradation.

## Results

**REX1 is the prime target of RNF12 in ESCs.** We previously performed an immunoprecipitation of RNF12 and identified REX1 as an RNF12 interaction partner, which is ubiquitinated by RNF12 to be targeted for degradation[11,16]. To identify the full spectrum of RNF12 targets in ESCs, we performed quantitative proteomics by stable isotope labelling of amino acids in cell culture (SILAC) and compared protein extracts from $Rnf12^{-/-}$ and wild type (WT) ESCs (Supplementary Fig. 1a; Supplementary Data 1). This analysis revealed REX1 to be the protein with the strongest increase in stability in extracts from $Rnf12^{-/-}$ cells, as compared to WT cells (Fig. 1a; Supplementary Fig. 1b). This indicates that REX1 is the main target of RNF12 for proteasomal degradation in ESCs. We also compared extracts of WT ESCs cultured in the presence or absence of the proteasome inhibitor MG132 (Supplementary Fig. 1c). REX1 and RNF12 were found to be among the proteins with the largest change in abundance in ESCs upon addition of the proteasome inhibitor MG132 (Supplementary Fig. 1d, e; Supplementary Data 1) highlighting their high turnover in ESCs.

***Rex1* deletion rescues the rXCI phenotype of $Rnf12^{-/-}$ ESCs.** As REX1 appears to be the primary substrate of RNF12 in ESCs, genetic removal of REX1 from $Rnf12^{-/-}$ ESCs may complement their XCI phenotype. To address this question, we firstly generated $Rnf12^{CR-/CR-}$ ESC lines by CRISPR/Cas9-mediated removal of the complete open reading frame of *Rnf12* in F1 129/Sv:Cast/EiJ (129:cas) ESCs (Supplementary Fig. 2a, b), and compared them to our previously generated $Rnf12^{-/-}$ ESCs[17] which still express the N-terminal 333 amino acids of RNF12, encoding the nuclear localization signal and part of the basic domain but excluding the catalytic Ring finger domain. Targeting and loss of RNF12 in $Rnf12^{CR-/CR-}$ ESCs was confirmed by PCR analysis on genomic DNA and western blotting (WB) analysis (Fig. 1b, c; Supplementary Fig. 3a). As expected, a marked increase of REX1 protein levels in $Rnf12^{CR-/CR-}$ and $Rnf12^{-/-}$ ESCs was observed by WB analysis (Fig. 1c; Supplementary Fig. 3a). Accordingly, we observed by immunofluorescence (IF) staining that increased REX1 expression is nuclear in both $Rnf12^{CR-/CR-}$ and $Rnf12^{-/-}$ ESCs (Fig. 1d), and in contrast to the homogeneous OCT4-staining, REX1 expression is heterogeneous within individual ESC colonies and overlaps with cells displaying high NANOG expression as previously described[18] (Supplementary Fig. 3b). This indicates that the absence of functional RNF12 causes nuclear accumulation of REX1. Quantitative RT-PCR and *Xist* RNA-FISH analysis indicated that *Xist* expression and *Xist* coating of the Xi was severely compromised in differentiating $Rnf12^{CR-/CR-}$ ESCs, both in monolayer and embryoid body (EB) differentiating conditions, confirming our previous observations that *Rnf12* is required for rXCI in vitro[17] (Supplementary Fig. 3c-g). We then generated $Rnf12^{CR-/CR-}:Rex1^{+/CR-}$ and $Rnf12^{CR-/CR-}:Rex1^{CR-/CR-}$ ESC lines by CRISPR/Cas9-mediated deletion of most of the open reading frame of *Rex1* (Supplementary Fig. 2a, b). Targeting was confirmed by PCR analysis on genomic DNA (Fig. 1b). WB and RT-qPCR analysis confirmed loss of *Rnf12* and *Rex1* expression in $Rnf12^{CR-/CR-}:Rex1^{CR-/CR-}$ double-knockout (DKO) ESCs (Fig. 1c; Supplementary Fig. 4a, b). REX1 protein levels were also increased and accumulated in the nucleus in the absence of RNF12 in $Rnf12^{CR-/CR-}:Rex1^{+/CR-}$ ESCs (Fig. 1c; Supplementary Fig. 4c). rXCI was rescued in $Rnf12^{CR-/CR-}:Rex1^{CR-/CR-}$ but not in $Rnf12^{CR-/CR-}:Rex1^{+/CR-}$ differentiating ESCs (Fig. 1e–g; Supplementary Fig. 4d,e). A slight delay in rXCI in $Rnf12^{CR-/CR-}:Rex1^{CR-/CR-}$ ESCs was observed compared to WT ESCs, which is likely related to the slower differentiation kinetics observed in

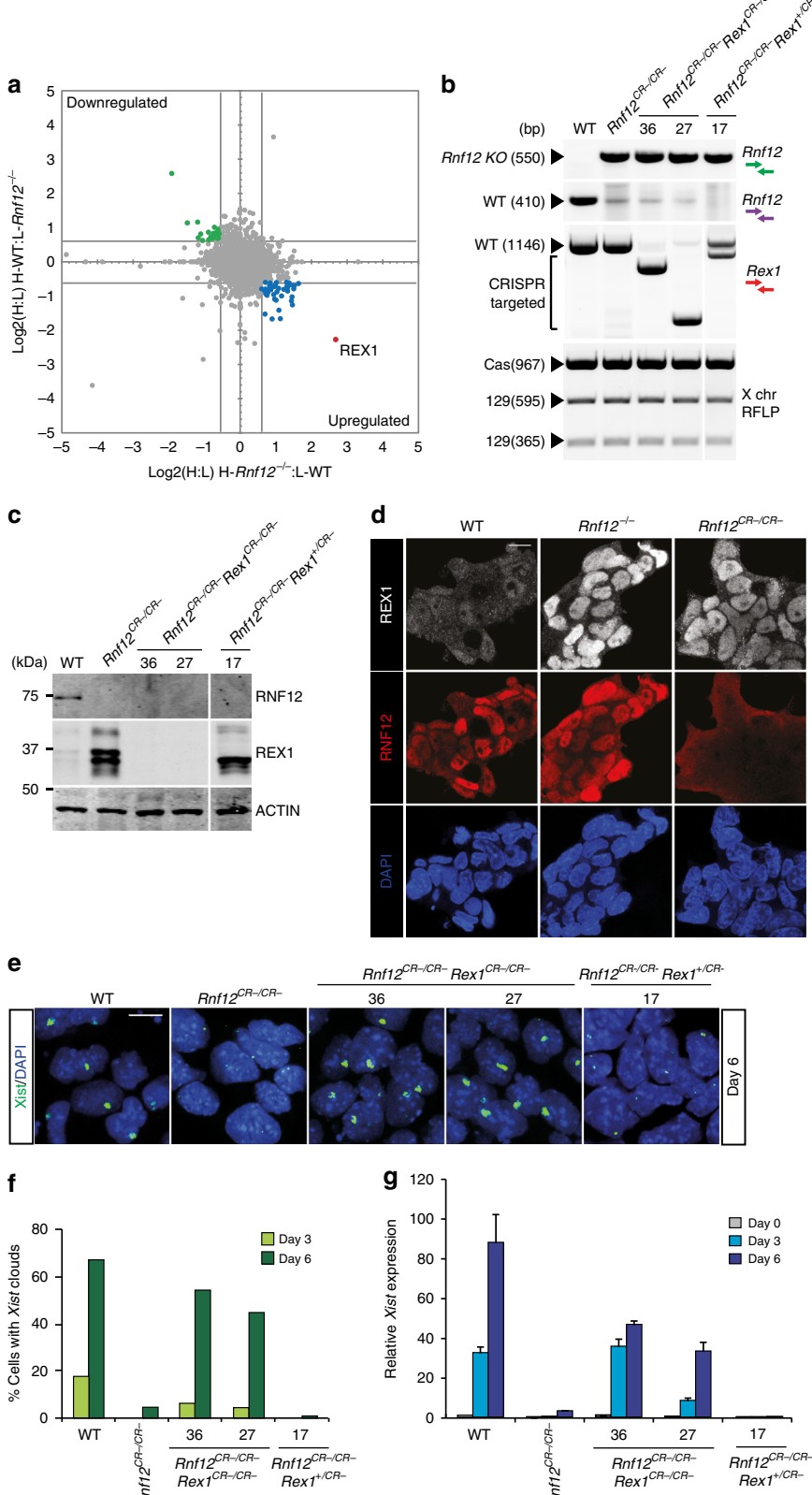

$Rnf12^{CR-/CR-}{:}Rex1^{CR-/CR-}$ ESCs (Supplementary Fig. 4f), and not due to a defect in rXCI (Supplementary Fig. 4g). These results illustrate the crucial role for RNF12-mediated degradation of REX1 in the initiation of rXCI in vitro.

**REX1 is dispensable for XCR, iXCI and rXCI.** *Rex1* is expressed at all stages during mouse pre-implantation development, where iXCI takes place (2- to 4-cell stage and trophoblast), and in cell types where the Xi is reactivated in the mouse life cycle: in the

**Fig. 1** Abrogation of rXCI in *Rnf12*$^{-/-}$ differentiating ESCs is rescued by knockout of *Rex1*. **a** Scatter plot showing the correlation of the H:L log2 ratios for the quantitatively identified proteins in the SILAC experiment in ESCs between two biological replicates (heavy-*Rnf12*$^{-/-}$:light-WT and heavy-WT:light-*Rnf12*$^{-/-}$) showing RNF12-dependent changes in REX1 stability. Upregulated proteins are indicated in blue (log2 ratios >0.585 and log2 ratios <−0.585 on the *x* and *y* axes, respectively). Depleted proteins are indicated in green (log2 ratios <−0.585 and log2 ratios >0.585 on the *x* and *y* axes, respectively). **b** Genotyping analysis of *Rnf12* and *Rex1* deletions in WT, *Rnf12*$^{CR-/CR-}$, *Rnf12*$^{CR-/CR-}$:*Rex*$^{CR-/CR-}$ (clones 36 and 27) and *Rnf12*$^{CR-/CR-}$:*Rex*$^{+/CR-}$ (clone 17) ESCs. Bottom panel shows a Pf1M1 restriction fragment length polymorphism (RFLP) analysis to detect the presence of two X chromosomes in the same ESC lines. Location of green, purple and red genotyping primers is depicted in Supplementary Fig. 2b. **c** Nuclear extracts of WT, *Rnf12*$^{CR-/CR-}$, *Rnf12*$^{CR-/CR-}$:*Rex*$^{CR-/CR-}$ (clones 36 and 27) and *Rnf12*$^{CR-/CR-}$:*Rex*$^{+/CR-}$ (clone 17) ESCs were immunoblotted with RNF12 and REX1 antibodies. ACTIN was used as a loading control. Uncropped WB images are found in Supplementary Fig. 10a. **d** Immunohistochemistry of REX1 (grey), RNF12 (red) and DNA (DAPI, blue) of WT, *Rnf12*$^{-/-}$ and *Rnf12*$^{CR-/CR-}$ ESCs. Note the recognition of the remaining 333 aa in RNF12$^{-/-}$ ESCs by the RNF12 antibody. Scale bar: 20 μm. **e** *Xist* RNA-FISH (FITC) analysis on WT, *Rnf12*$^{CR-/CR-}$, *Rnf12*$^{CR-/CR-}$:*Rex*$^{CR-/CR-}$ (clones 36 and 27) and *Rnf12*$^{CR-/CR-}$:*Rex*$^{+/CR-}$ (clone 17) ESCs at day 6 of differentiation. DNA was stained with DAPI (blue). Scale bar: 20 μm. **f** Quantification of cells with *Xist* clouds in WT, *Rnf12*$^{CR-/CR-}$, *Rnf12*$^{CR-/CR-}$:*Rex*$^{CR-/CR-}$ (clones 36 and 27) and *Rnf12*$^{CR-/CR-}$:*Rex*$^{+/CR-}$ (clone 17) ESCs at day 3 and day 6 of differentiation. **g** QPCR analysis of *Xist* expression in undifferentiated, day 3 and day 6 differentiated WT, *Rnf12*$^{CR-/CR-}$, *Rnf12*$^{CR-/CR-}$:*Rex*$^{CR-/CR-}$ (clones 36 and 27) and *Rnf12*$^{CR-/CR-}$:*Rex*$^{+/CR-}$ (clone 17) ESCs (average expression ± s.d., *n* = 3 biological replicates)

epiblast lineage of E4.5 female blastocysts as well as in developing PGCs, at E10.5[14,19,20]. Although *Rex1* mutant mice were reported to be viable, they are born at sub-Mendelian ratios and display defects in imprinted gene regulation[12,13]. In rXCI, REX1 is an important repressor of *Xist*[11] and its expression in the ICM and PGCs makes it a candidate factor in X reactivation.

To investigate the effect of *Rex1* ablation on XCI, we generated *Rex1* knockout (KO) mice by blastocyst injection of *Rex1*$^{+/-}$ F1 129:cas ESCs generated by gene targeting through homologous recombination (Fig. 2a–c, Supplementary Fig. 2a). *Rex1* KO mice were backcrossed for at least six generations in two different genetic backgrounds (129/Sv and Cast/EiJ). In agreement with previous studies, we found a reduced litter size for *Rex1*$^{-/-}$ crosses, and no significant gender bias against birth of female animals (Fig. 2d). We subsequently isolated blastocysts and established a *Rex1*$^{-/-}$ F1 129:cas female ESC line. In ESC monolayer differentiation experiments, we observed by quantitative RT-PCR and *Xist* RNA-FISH analysis increased *Xist* expression and reduced *Tsix* expression (Supplementary Fig. 5a–c) in *Rex1*$^{+/-}$ and *Rex1*$^{-/-}$ ESCs, with significantly more *Xist* clouds in *Rex1*$^{+/-}$ and *Rex1*$^{-/-}$ ESCs than in WT controls (Fig. 2e, f). More importantly, we also observed significantly more cells with two *Xist* clouds (Fig. 2e, f), indicating an important role for REX1 in repression of *Xist*, providing the feedback required to prevent XCI of too many X chromosomes.

To test the involvement of *Rex1* in the reactivation of the Xi in the ICM or PGCs, we firstly analysed the ICM of E3.5 embryos and epiblast of E4.5 embryos by IF detecting H3K27me3 marking the Xi[21,22], together with OCT4 and KLF4, which stain the cells specific to the ICM and epiblast, respectively (Fig. 3a, b; Supplementary Fig. 6a). This revealed no delay in timing of loss of the H3K27me3-coated Xi between *Rex1*$^{-/-}$ and WT female embryos, indicating that *Rex1* is dispensable for Xi reactivation in the ICM. To confirm these results, we crossed our *Rex1*$^{-/-}$ female mice with *Rex1*$^{-/-}$ male mice containing an X-linked GFP reporter adjacent to Hprt[23] (*Rex1*$^{-/-}$:Hprt$^{GFP/y}$). Comparison of *Rex1*$^{-/-}$:Hprt$^{+/GFP}$ to Hprt$^{+/GFP}$ E4.5 female blastocysts showed no difference in the amount of Xp-reactivated GFP-positive cells lacking the H3K27me3 domain (Supplementary Fig. 6b, c), confirming our previous results. Analysis of XCR in PGCs by H3K27me3 IF staining together with OCT4 to map them revealed no difference in the rate of XCR between *Rex1*$^{-/-}$ and WT E9.5 and E11.5 female embryos (Fig. 3c, d; Supplementary Fig. 6d, e). Together, these results show that REX1 is not required for the reactivation of the Xi chromosome in vivo.

We then investigated the role of REX1 in iXCI and rXCI in vivo. *Xist* RNA-FISH analysis of *Rex1*$^{-/-}$ blastocyst outgrowths showed no differences in iXCI compared to WT cells

(Supplementary Fig. 7a, b). Allele-specific expression analysis in *Rex1*$^{-/-}$ and WT E11.5 female embryos of X-linked genes *Xist*, *G6pdx* and *Mecp2* indicated normal iXCI and rXCI, with preferential inactivation of the paternally inherited X (cas) in the VYSE of F1 129:cas embryos, and rXCI in embryonic tissues (Fig. 3e, f; Supplementary Fig. 7c). rXCI is skewed in 129:cas embryos due to the presence of two distinct genetic X choosing elements in the F1 genetic background leading to preferential inactivation of the 129 X chromosome (70/30% 129/cas)[24]. Examination of rXCI in five different organs of 4-week-old female mice was in line with the above findings, revealing no effect of heterozygous and homozygous *Rex1* mutations on the allelic origin of *Xist*, *G6pdx* and *Mecp2* expression (Fig. 3g, h; Supplementary Fig. 7d, e). This indicates that repression of *Xist* and XCR in the ICM and PGCs happen in the absence of REX1 suggesting that *Rex1* is dispensable for iXCI and rXCI regulation in vivo.

**Rnf12$^{-/-}$ embryos accumulate REX1 and lack iXCI.** To study the role of REX1 in the lethality of maternally transmitted mutant RNF12 alleles[7], we generated *Rnf12* KO mice from *Rnf12*$^{-/-}$ F1 129:cas ESCs[17]. Litters of crosses between *Rnf12*$^{+/-}$ females with WT males were analysed in a C57BL/6 background. We observed no viable female mice with a maternal *Rnf12* deletion in the offspring of 30 breedings, whereas female mice with a paternal *Rnf12* deletion were born at expected Mendelian ratios, confirming the reported crucial role for *Rnf12* in iXCI[7] (Fig. 4a). *Rnf12*$^{-/y}$ males displayed partial lethality before genotyping at 5 dpn, implying XCI-independent effects of the *Rnf12* deletion (Fig. 4a). These differences were not C57BL/6-specific since we obtained the same results in a Cast/EiJ background (Supplementary Fig. 8a). Interestingly, REX1 staining revealed that the observed loss of iXCI in *Rnf12*$^{-/-}$ embryos indicated by the absence of H3K27me3 domains representing the Xi, coincided with strongly increased REX1 protein levels and nuclear localization in the epiblast and trophectoderm of E4.5 female embryos (Fig. 4b). On the contrary, *Rnf12*$^{-/+}$ embryos showed REX1 protein levels similar to WT embryos, and a certain degree of iXCI as seen by the presence of a small number of cells with H3K27me3 domains in some embryos (Supplementary Fig. 8b). The almost absence of iXCI in *Rnf12*$^{-/+}$ trophectoderm cells is presumably due to direct repression of *Xist*, instead of activation of the paternal *Tsix* allele by REX1 since we did not observe any paternal (cas) *Tsix* expression in these blastocysts (Supplementary Fig. 8c). These results indicate that in WT embryos, RNF12 controls REX1 protein levels in extraembryonic tissues and suggests that the loss of iXCI in *Rnf12* mutant embryos may be related to increased levels of REX1.

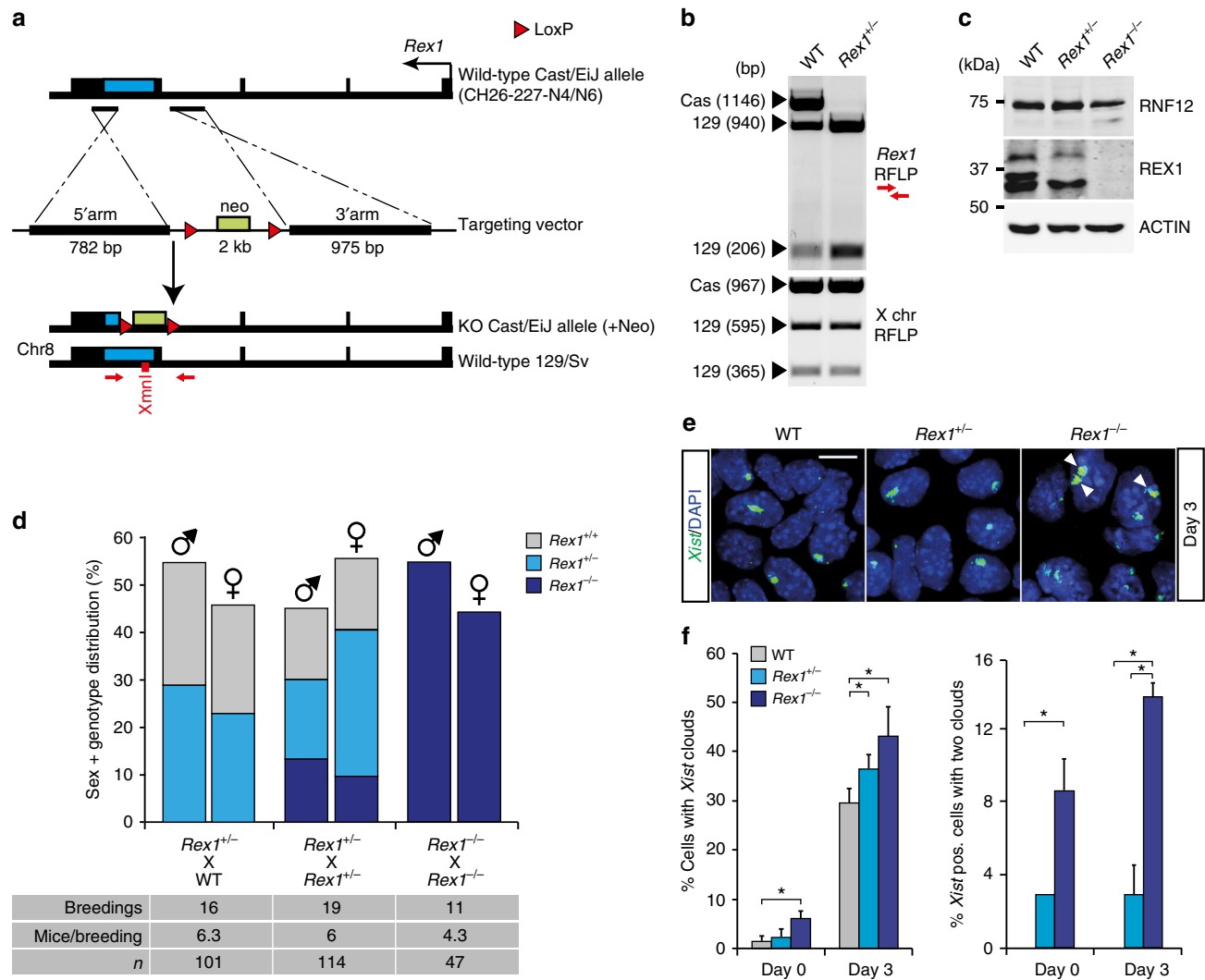

**Fig. 2** *Rex1* knockout mice are viable and fertile. **a** BAC targeting strategy to generate the *Rex1* knockout cas allele in 129:cas ESCs. The *Rex1* coding sequence is shown as a blue box. The start site is indicated by a black arrow. LoxP sites, denoted as red triangles, flank the neomycin-resistance gene (neo) as a positive gene selection marker. Primers used to validate gene recombination in ESCs and to genotype are shown as red arrows. **b** Validation of *Rex1* knockout cas allele recombination by XmnI RFLP analysis of WT and *Rex1*-targeted ESCs (top panel). Pf1M1 RFLP analysis to detect presence of two X chromosomes (bottom panel). **c** Nuclear extracts of WT, *Rex1*$^{+/-}$ and *Rex1*$^{-/-}$ ESCs were immunoblotted with RNF12 and REX1 antibodies. ACTIN was used as a loading control. Uncropped WB images are found in Supplementary Fig. 10b. **d** Sex and genotype distribution from different matings of *Rex1*-deficient mice. Number of breedings, number of mice per breeding and total number of mice are indicated below. No significant bias against the birth of female animals was observed ($\chi^2$ test, $p > 0.05$). **e** *Xist* RNA-FISH (FITC) analysis on WT, *Rex1*$^{+/-}$ and *Rex1*$^{-/-}$ ESCs at day 3 of differentiation. DNA was stained with DAPI (blue). White arrows indicate the presence of two clouds within a nucleus. Scale bar: 20 μm. **f** Quantification of *Xist*-positive cells (left panel) and *Xist*-positive cells with two clouds (right panel) in WT, *Rex1*$^{+/-}$ and *Rex1*$^{-/-}$ ESCs at day 0 and day 3 of differentiation. Asterisks indicate $p$-value <0.05, two-tailed Student's $t$ test (average expression ± s.d., $n = 1$–3 biological replicates)

***Rex1* deletion rescues the lethality of *Rnf12*$^{-/-}$ mice.** To test whether stabilization of REX1 in *Rnf12* mutant embryos might be related to the lethality associated with maternal transmission of *Rnf12* mutant alleles, we crossed our *Rnf12* KO mice and *Rex1* KO mice to generate *Rnf12*$^{-/-}$:*Rex1*$^{-/-}$ DKO mice. In contrast to *Rnf12*$^{-/+}$ or *Rnf12*$^{-/-}$ mice that were never obtained in a *Rex1* WT or heterozygous background, several *Rnf12*$^{-/+}$:*Rex1*$^{-/-}$ and *Rnf12*$^{-/-}$:*Rex1*$^{-/-}$ female mice were born (Fig. 5a; Supplementary Fig. 9a). In general, litters were smaller, but no gender or allele bias was observed (Fig. 5a). We confirmed these results by crossing *Rnf12*$^{+/-}$:*Rex1*$^{-/-}$ or *Rnf12*$^{-/-}$:*Rex1*$^{-/-}$ females with *Rex1*$^{-/+}$ males, whose *Rnf12*$^{-/+}$:*Rex1*$^{-/-}$ daughters were viable but their *Rnf12*$^{-/+}$:*Rex1*$^{-/+}$ sisters were not (Supplementary Fig. 9a). IF staining on *Rnf12*$^{-/-}$:*Rex1*$^{-/-}$ E4.5 female blastocysts from compound homozygous crosses confirmed loss of REX1 and

showed proper XCR in the epiblast during pre-implantation development, as seen by the loss of H3K27me3 domains corresponding to the Xi (Fig. 5b). Also, IF staining detecting H3K27me3 and OCT4 revealed normal XCR in PGCs (Supplementary Fig. 9b, c), suggesting that reactivation of the Xi is normal in *Rnf12*$^{-/-}$:*Rex1*$^{-/-}$ female blastocysts and embryos.

We then investigated iXCI and rXCI in *Rnf12*$^{-/-}$:*Rex1*$^{-/-}$ blastocysts, embryos and adults. REX1 and H3K27me3 IF staining of *Rnf12*$^{-/-}$:*Rex1*$^{-/-}$ blastocysts showed normal iXCI (H3K27me3 domains corresponding to the Xi) compared to WT blastocysts (Fig. 5b), in line with *Xist* RNA-FISH analysis in trophoblast cells of *Rnf12*$^{-/-}$:*Rex1*$^{-/-}$ blastocyst outgrowths (Supplementary Fig. 7a, b), indicating that iXCI in *Rnf12*$^{-/-}$:*Rex1*$^{-/-}$ blastocysts is normal. Allele-specific RT-PCR analysis examining *Xist*, *G6pdx* and *Mecp2* expression on RNA isolated

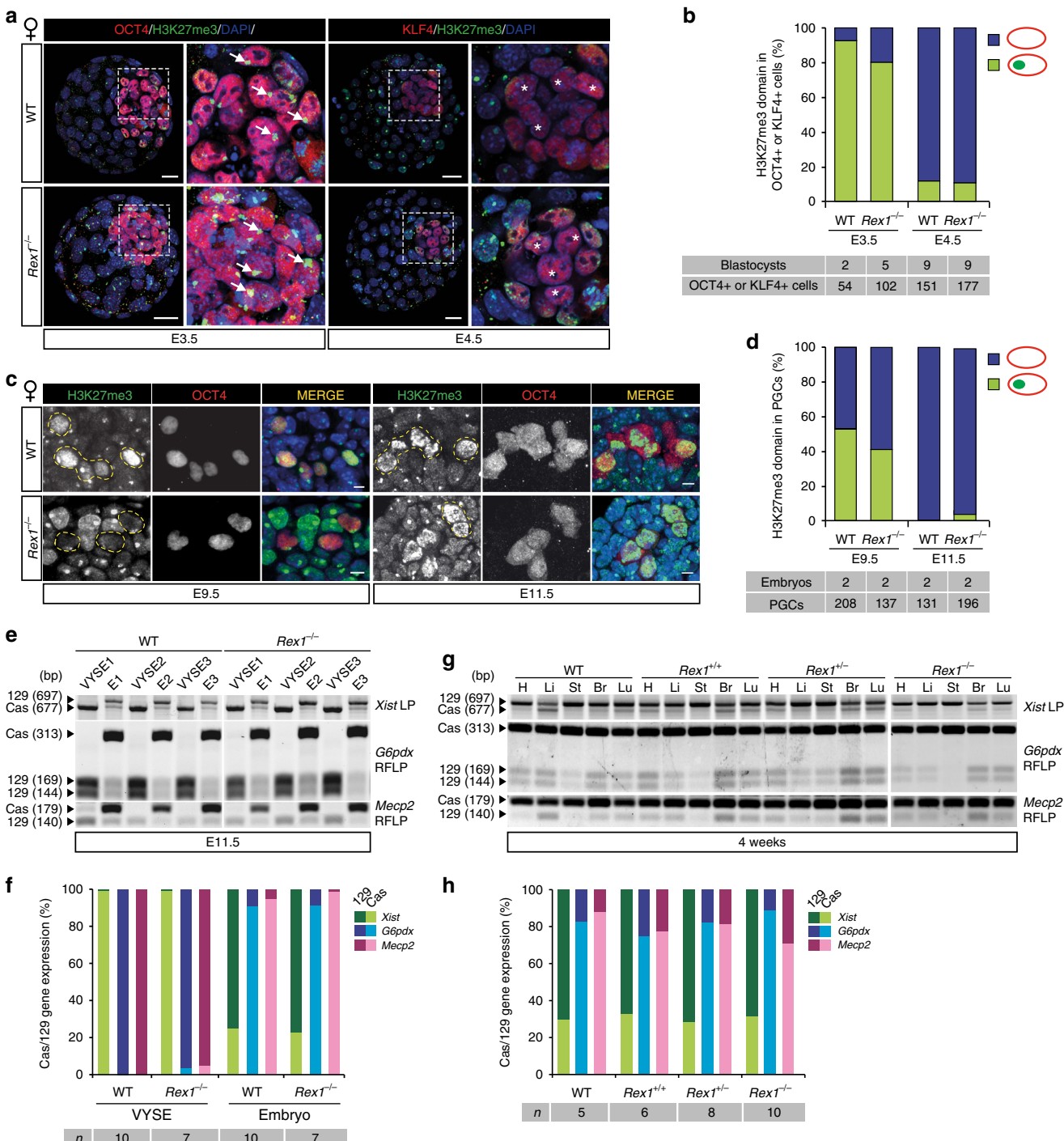

**Fig. 3** XCR and imprinted and random XCI in female *Rex1⁻/⁻* mice are not compromised. **a** Representative *Z*-stack projections of WT and *Rex1⁻/⁻* E3.5 and E4.5 female blastocysts immunostained for H3K27me3 (Xi marker, green), the lineage markers OCT4 (E3.5 ICM, red, left panels) and KLF4 (E4.5 epiblast, red, right panels) and DNA (DAPI, blue). Whole embryo and ICM/epiblast higher magnification for each embryo are shown (white boxes). Arrowheads mark the Xi in OCT4+ cells. KLF4+ cells display XCR (stars). Scale bars: 20 μm. **b** Quantification of H3K27me3 domains (green) in E3.5 and E4.5 ICMs of WT and *Rex1⁻/⁻* embryos in **a**. **c** Representative paraffin sections of female WT and *Rex1⁻/⁻* E9.5 embryo hindguts and E11.5 embryo trunks immunostained for OCT4 (PGC marker, red) and H3K27me3 (Xi marker, green). H3K27me3 domains are present in somatic cells and in some E9.5 PGCs, while they are lost in E11.5 PGCs (XCR). Representative PGCs are marked with yellow dashed lines. Scale bars: 5 μm. **d** Quantification of cells with an H3K27me3 domain in female WT and *Rex1⁻/⁻* PGCs at E9.5 and E11.5. Number of embryos and PGCs analysed are indicated. **e** *Xist*, *G6pdx* and *Mecp2* allele-specific RNA expression analysis in E11.5 WT and *Rex1⁻/⁻* female embryos (E) and corresponding VYSE. LP, length polymorphism. **f** Quantification of the average allelic *Xist* (green), *G6pdx* (blue) and *Mecp2* (pink) expression in WT and *Rex1⁻/⁻* embryos and VYSE in **e**. Light/dark colours indicate cas or 129 allelic origin, respectively. The number of mice analysed is indicated. **g** Representative *Xist*, *G6pdx* and *Mecp2* allele-specific RNA expression analysis in heart (H), liver (Li), stomach (St), brain (Br) and lung (Lu) of WT, *Rex1⁺/⁺* (from a heterozygous cross), *Rex1⁺/⁻* and *Rex1⁻/⁻* 4-week-old mice. **h** Quantification of the average allelic *Xist* (green), *G6pdx* (blue) and *Mecp2* (pink) expression in several WT, *Rex1⁺/⁺* (from a heterozygous cross), *Rex1⁺/⁻* and *Rex1⁻/⁻* 4-week-old mice in **g** and additional mice. Light/dark colours indicate cas or 129 allelic origin, respectively. The number of mice analysed is indicated

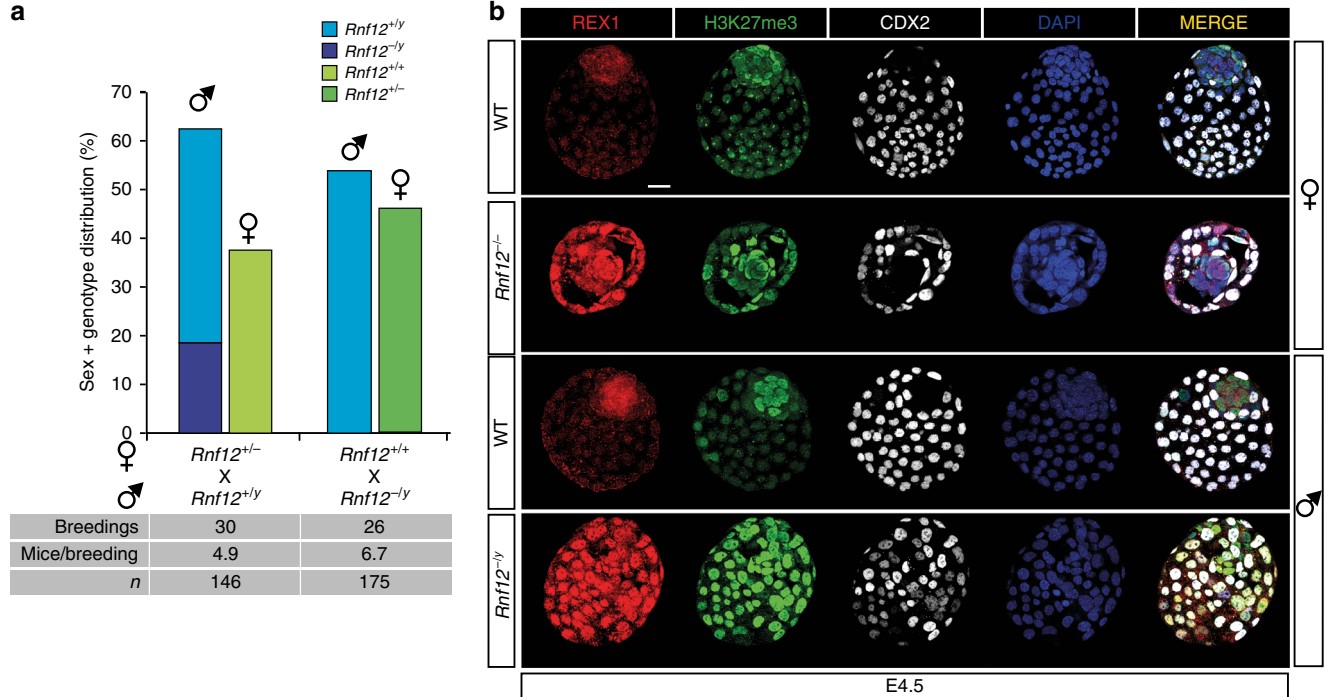

**Fig. 4** Rnf12 KO embryos show REX1 stabilization in embryonic and extraembryonic tissues. **a** Sex and genotype distribution from different matings of *Rnf12*-deficient mice in a C57BL/6 background. Number of breedings, number of mice per breeding and total number of mice are indicated. Note that no female embryos were born with a maternally transmitted *Rnf12* deleted allele. No significant differences were observed between the number of females with a paternal mutant allele and their WT brothers ($\chi^2$ test, $p > 0.05$). A significant slight lethality associated with the mutation in male mice compared to WT brothers was observed ($\chi^2$ test, $p = 1.05E{-}4$). **b** Representative Z-stack projections of WT, *Rnf12*$^{-/-}$ and *Rnf12*$^{-/y}$ E4.5 blastocysts immunostained for REX1 (red), H3K27me3 (Xi marker, green), the trophectoderm marker CDX2 (grey) and DNA (DAPI, blue). Scale bars: 20 μm

from E11.5 *Rnf12*$^{-/+}$:*Rex1*$^{-/-}$ extraembryonic VYSE and embryos confirmed normal initiation of iXCI and rXCI similar to WT embryos, with preferential expression of *Xist* from the paternal cas X chromosome in VYSE, and skewed rXCI towards inactivation of the 129 X chromosome observed in the embryo (Fig. 5c, d; Supplementary Fig. 9d). *Rnf12*$^{-/+}$:*Rex1*$^{-/-}$ and *Rnf12*$^{-/-}$:*Rex1*$^{-/-}$ adults showed no differences in rXCI with WT adults (Fig. 5e, f; Supplementary Fig. 9e). These results show that the lethality due to the lack of iXCI in RNF12 KO mice can be fully rescued by knocking out *Rex1*, indicating that RNF12-mediated targeting of REX1 is crucial for proper regulation of iXCI.

## Discussion

The intricate relationship between REX1 and RNF12 during embryonic development is intriguing. REX1 is expressed during all stages of pre-implantation development when iXCI takes place, and decreases rapidly before rXCI initiates in epiblast cells soon after implantation[14,19]. Although *Rex1*$^{-/-}$ litters are smaller, possibly due to placental imprinting problems[12], no sex-bias was observed in our study. The absence of *Rex1* did not affect iXCI, nor XCR, which does not completely rule out a role for *Rex1* in XCR, as this process might be regulated by redundant mechanisms possibly including *Prdm14* and other putative *Xist* regulators[21].

Our studies indicate that REX1 is the main target of RNF12 in the initiation of iXCI in the mouse. It has been previously shown that RNF12 removal leads to absence of *Xist* clouds in trophoblast cells, resulting in lack of iXCI and embryonic lethality[7]. We propose that this absence of *Xist* clouds is caused by increased REX1 levels in the *Rnf12*$^{-/-}$ pre-implantation embryo. Lethality

associated with maternal inheritance of an *Rnf12* mutant allele is bypassed by removing *Rex1*. In this line, at least one copy of *Rnf12* needs to remain active for the iXCI process to proceed by preventing accumulation of REX1 and subsequent silencing of *Xist* on the paternal X[7], similar to our findings in rXCI in ESCs[10,11]. We propose a model where the *Rnf12-Rex1* axis regulates initiation of iXCI and the maintenance of the Xi in the pre-implantation embryo (Fig. 6). During pre-implantation development, both *Rex1* and *Rnf12* are expressed. If the *Rnf12* mutant allele is maternally transmitted, *Rnf12*$^{-/+}$ cells in the pre-implantation embryo will initiate inactivation of their paternal WT X chromosome, effectively leading to an *Rnf12* homozygous knock out situation. This results in transient accumulation of REX1 protein, not detectable by IF staining, but high enough to likely repress *Xist* expression from the paternal X, analogous to its action in preventing rXCI in ESCs[11]. A small number of cells will escape this situation and inactivate the paternal X chromosome but the general lack of paternal Xi will eventually cause the death of the embryo. If *Rex1* is genetically removed, *Xist* expression can no longer be repressed, allowing the inactivation of the paternal X. Maternal transmission of an *Rnf12* mutant allele is in this way no longer lethal as iXCI is rescued.

Overexpression of *Rnf12* has also been shown to facilitate erasure of the repressive imprint on the maternally inherited *Xist* allele in XmXm parthenogenetic embryos[25]. This effect was eliminated by collective knockdown of *Rnf12* and *Rex1*, indicating that REX1 is normally involved in maintenance of *Xist* repression on the Xm. These and our results highlight the fact that REX1 expression needs to be precisely titrated. Too much REX1 results in repression of *Xist* on the Xp and embryonic lethality, whereas too little results in de-repression of *Xist* in an XmXm partheno-genetic setting. Interestingly, *Rex1* KO embryos with bi-parental

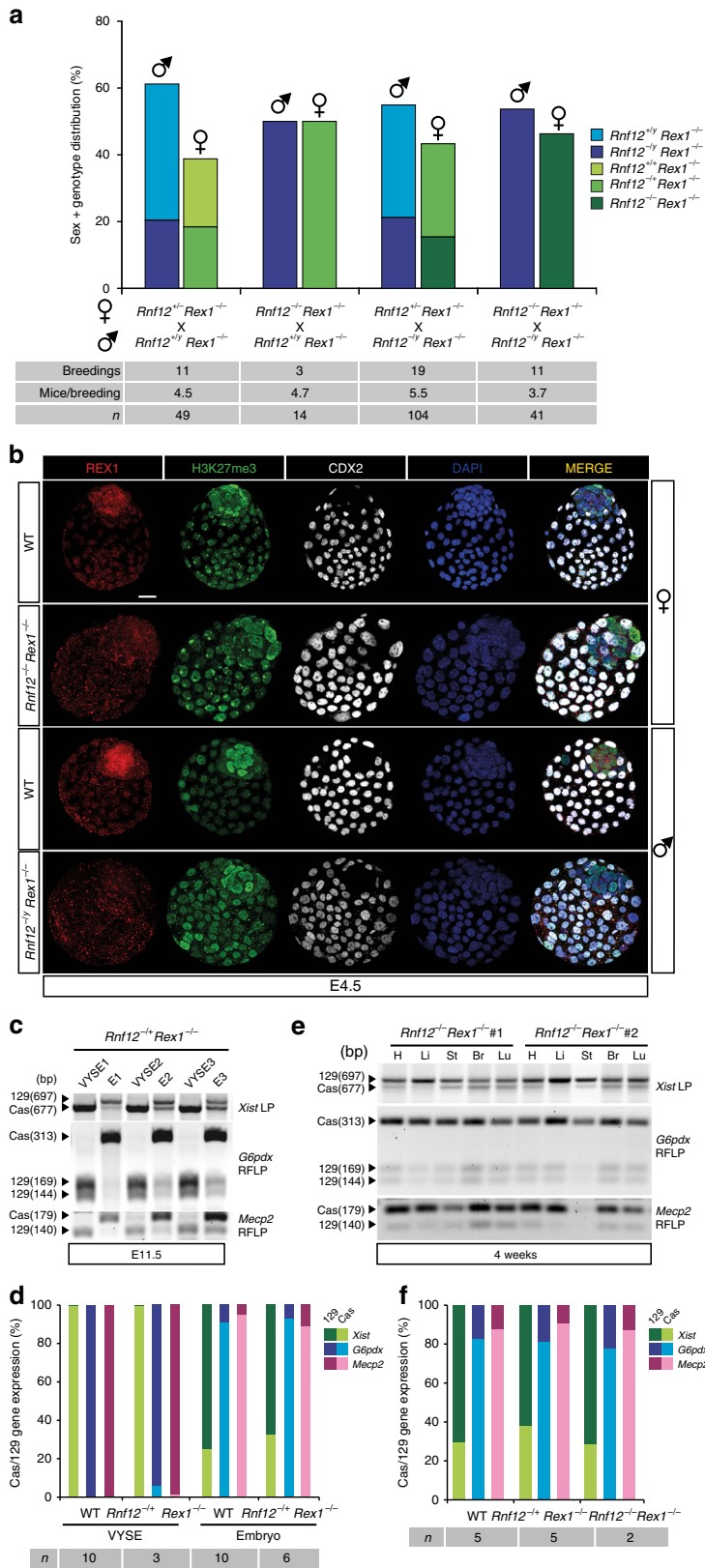

Xm and Xp chromosomes do not display any iXCI defect indicating that additional mechanisms are in place to facilitate maintenance of iXCI during embryonic development in the absence of REX1. During rXCI, the Xist 5′ regulatory region containing YY1 binding sites becomes asymmetrically methylated[26]. Mono-allelic methylation prevents YY1 binding and activation of that Xist allele, while binding competition of YY1 and REX1 to the unmethylated allele results in activation and repression of Xist, respectively. In iXCI, a similar mechanism might be in place to facilitate Xist expression from the paternal X,

**Fig. 5** $Rnf12^{-/-}:Rex1^{-/-}$ double knockout mice are viable and have normal iXCI and rXCI. **a** Sex and genotype distribution from different $Rnf12$ mutant crossings in an $Rex1$-deficient background. Number of breedings, number of mice per breeding and total number of mice are indicated. Note that female embryos were born with a maternally transmitted $Rnf12$ deleted allele in an $Rex1^{-/-}$ background. No significant bias in gender or genotype was observed ($\chi^2$, $p > 0.05$). **b** Representative Z-stack projections of WT, $Rnf12^{-/-}:Rex1^{-/-}$ and $Rnf12^{-/y}:Rex1^{-/y}$ E4.5 blastocysts immunostained for REX1 (red), H3K27me3 (Xi marker, green), the trophectoderm marker CDX2 (grey) and DNA (DAPI, blue). Scale bars: 20 μm. WT samples are same control samples as in Fig. 4b. **c** Xist, G6pdx and Mecp2 allele-specific RNA expression analysis in E11.5 female $Rnf12^{-/+}:Rex1^{-/-}$ embryos (E) and corresponding VYSE. **d** Quantification of the average allelic Xist (green), G6pdx (blue) and Mecp2 (pink) expression in E11.5 female WT and $Rnf12^{-/+}:Rex1^{-/-}$ embryos and VYSE in **c**. Light/dark colours indicate cas or 129 allelic origin, respectively. WT samples are same control samples as in Fig. 3f. The number of mice analysed is indicated. **e** Xist, G6pdx and Mecp2 allele-specific RNA expression analyses in heart (H), liver (Li), stomach (St), brain (Br) and lung (Lu) in two $Rnf12^{-/-}:Rex1^{-/-}$ 4-week-old female mice. **f** Quantification of the average allelic Xist (green), G6pdx (blue) and Mecp2 (pink) expression in WT and $Rnf12^{-/+}:Rex1^{-/-}$ and $Rnf12^{-/-}:Rex1^{-/-}$ 4-week-old female mice in **e**. Light/dark colours indicate cas or 129 origin, respectively. WT samples are the same control samples as in Fig. 3h. The number of mice analysed is indicated

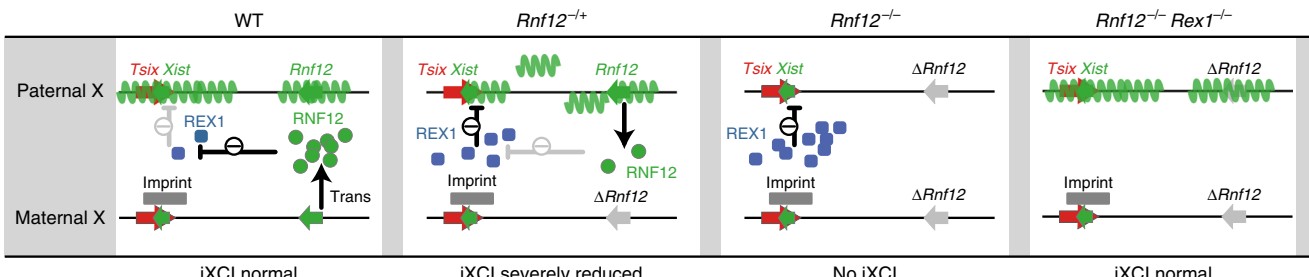

**Fig. 6** Model for iXCI regulation by the $Rex1$-$Rnf12$ axis. In a WT background, REX1 is expressed during the early stages after fertilization, but is degraded by RNF12. This leads to the upregulation of the paternal Xist allele, whilst its maternal counterpart is prevented from upregulation by an imprint. In $Rnf12^{-/+}$ embryos, the RNF12 level is sufficient to prevent REX1 stabilization, allowing for paternal Xist upregulation. However, this will result in inactivation of the paternal $Rnf12$ copy, which leads to an RNF12 KO situation, in its turn resulting in brief REX1 stabilization and preventing further paternal Xist upregulation. This feedback loop will prevent proper iXCI in trophoblast cells leading to $Rnf12^{-/+}$ embryo death. In $Rnf12^{-/-}$ embryos, in the absence of RNF12, REX1 accumulates, preventing upregulation of Xist from the paternal allele. iXCI is absent and the embryos die. In $Rnf12^{-/-}:Rex1^{-/-}$ embryos, the entire $Rex1$-$Rnf12$ axis is absent and iXCI can proceed normally as in WT embryos

in conjunction with a repressive imprint on the maternal Xist allele. Accumulation of REX1 in the absence of RNF12 might outcompete the binding of YY1 and prevent the transcriptional activation of the paternal Xist.

Our present and previous studies indicate that in vitro, $Rnf12$-deficient ESCs show a complete loss of rXCI in a 129/Sv:Cast/EiJ hybrid genetic background[17]. For the present study, we generated $Rnf12^{CR-/CR-}$ ESCs through complete removal of the open reading frame of $Rnf12$, and compared these cells with $Rnf12^{-/-}$ ESCs that we generated in a previous study, which express a 333 aa N-terminal peptide that does not contain the catalytic Ring finger domain. Both $Rnf12^{CR-/CR-}$ ESCs and $Rnf12^{-/-}$ ESCs display stabilization and nuclear localization of REX1, contrasting a recent report that suggested a role for this 333 aa N-terminal peptide in the nuclear localization of the stabilized REX1[27]. Our quantitative mass spectrometry analysis revealed REX1 to be the main target of RNF12. Moreover, the loss of rXCI phenotype observed in differentiating $Rnf12^{CR-/CR-}$ KO ESCs is rescued in a compound $Rnf12:Rex1$ DKO background. This result indicates that REX1 accumulation instigated by the loss of $Rnf12$ is key to the absence of Xist upregulation. In addition, loss of REX1 leads to a significant increase of cells with two Xist clouds in differentiated $Rex1^{-/-}$ and $Rex1^{+/-}$ ESCs. These findings and the high turnover of these proteins, mediated by proteasomal degradation, provides a powerful feedback mechanism preventing XCI of both X chromosomes during ESC differentiation. In vivo, the role of the $Rnf12$-$Rex1$ axis in rXCI appears less prominent[8], presumably due to decreased expression levels of REX1 and RNF12 in the developing epiblast where rXCI is taking place[19], possibly allowing the embryos to overcome a REX1-mediated block of rXCI in $Rnf12^{-/-}$ epiblasts as was observed in tetraploid

complementation assays[8]. In addition, in embryos the regulation of rXCI seems more robust than in vitro, which might explain why we observe a subpopulation of $Rex1^{-/-}$ and $Rex1^{+/-}$ ESCs with two Xist clouds whereas in $Rex1^{-/-}$ and $Rex1^{+/-}$ embryos and adults rXCI appears unaffected. If XCI of two X chromosomes happens in the developing epiblast, we anticipate that these cells will quickly restore the Xa:Xa status to allow reinitiation of XCI, or will be counter selected and eliminated from the embryo. Intriguingly, the fact that $Rnf12^{-/-}:Rex1^{-/-}$ DKO mice are born, and that rXCI takes place in $Rnf12^{-/-}:Rex1^{-/-}$ DKO ESCs, highlights the existence of additional regulatory mechanisms that act in concert with the $Rnf12$-$Rex1$ axis to properly and timely execute XCI.

## Methods

**SILAC labelling of ESCs.** For the SILAC experiments, undifferentiated female WT line F1 2-1 and $Rnf12^{-/-}$ ESCs[17] were used. ESC lines were metabolically labelled by culturing either in "light" or "heavy" media for at least five passages (around 2 weeks), in order to achieve maximum incorporation of the isotope labelled amino acids into the proteins. For the protein stability experiment, cells were treated with either vehicle (dimethyl sulphoxide, DMSO) or proteasome inhibitor (15 μm MG132, Sigma, C2211) for 3 h prior to harvesting.

SILAC medium containing DMEM High Glucose (4.5 g l⁻¹) devoid of arginine and lysine (PAA Cell Culture Company) supplemented with 15% dialyzed foetal bovine serum (Invitrogen), 100 U ml⁻¹ penicillin, 100 μg ml⁻¹ streptomycin, 200 mM GlutaMAX (Invitrogen), 0.1 mM non-essential amino acids (NEAA), 1000 U ml⁻¹ LIF, 0.1 mM 2-mercaptoethanol (Sigma) and 200 mg l⁻¹ L-proline (Sigma, P0380) to avoid arginine-to-proline conversion[28]. Either naturally occurring isotopes of lysine and arginine (Arg0, Sigma, A5006; Lys0, Sigma, L5501) (light media) or the heavy isotopes (Arg10, Cambridge Isotope Laboratories, CNLM-539; Lys8, Cambridge Isotope Laboratories CNLM-291) (heavy media) were added to the medium at a concentration of 100 mg l⁻¹ for lysine and 40 mg l⁻¹ for arginine.

Medium was refreshed every day, and the cells were split every other day. The ESC lines were cultured for three passages on feeder cells and for an additional two

passages feeder-free, on 0.2% gelatin-coated cell culture plates. The SILAC media in the feeder-free culture was supplemented with 25 ng ml$^{-1}$ recombinant human bone morphogenic protein 4[29] (BMP4; PeproTech, 120-05). ESCs from both light and heavy cultures were harvested ($1 \times 10^8$ cells for each condition: WT and Rnf12$^{-/-}$ or WT in the presence or absence of the proteasome inhibitor) and mixed in a 1:1 ratio for subsequent nuclear protein extracts[11]. About 1 mg of protein from each extract was separated on a 4–12% NuPage Novex bis-Tris gel (Invitrogen, NP0321) and stained using the Colloidal Blue Staining kit (Invitrogen, LC6025) according to the manufacturer's instructions.

**Mass spectrometry.** SDS-PAGE gel lanes were cut into 2-mm slices and subjected to in-gel reduction with dithiothreitol, alkylation with iodoacetamide and digested with trypsin (sequencing grade; Promega). Nanoflow liquid chromatography tandem mass spectrometry (LC-MS/MS) was performed on an EASY-nLC coupled to a Q Exactive mass spectrometer (Thermo) or on an 1100 series capillary liquid chromatography system (Agilent Technologies) coupled to an LTQ-Orbitrap XL mass spectrometer (Thermo), both operating in positive mode. Peptides were trapped on a ReproSil C18 reversed phase column (Dr. Maisch; 1.5 cm × 100 μm) at a rate of 8 μl min$^{-1}$ and separated on a ReproSil-C18 reversed-phase column (Dr. Maisch; 15 cm × 50 μm) using a linear gradient of 0–80% acetonitrile (in 0.1% formic acid) during 120–170 min at a rate of ~200 nl min$^{-1}$ using a splitter (LTQ-Orbitrap XL) or not (Q Exactive). The elution was directly sprayed into the electrospray ionization (ESI) source of the mass spectrometer. Spectra were acquired in continuum mode; fragmentation of the peptides was performed in data-dependent mode using CID (LTQ-Orbitrap XL) or HCD (Q Exactive).

Raw mass spectrometry data were analysed with MaxQuant software (version 1.5.6.0) as described before[30]. A false discovery rate of 0.01 for proteins and peptides and a minimum peptide length of 6 amino acids were set. The Andromeda search engine[31] was used to search the MS/MS spectra against the Uniprot database (taxonomy: *Mus musculus*, release October 2016) concatenated with the reversed versions of all sequences. A maximum of two missed cleavages was allowed. The peptide tolerance was set to 10 ppm and the fragment ion tolerance was set to 0.6 Da for CID spectra and to 20 mmu for HCD spectra. The enzyme specificity was set to trypsin and cysteine. Carbamidomethylation was set as a fixed modification, while protein N-acetylation and GlyGly (K) were set as variable modifications. MaxQuant automatically quantified peptides and proteins based on standard SILAC settings (multiplicity = 2, K8R10). SILAC protein ratios were calculated as the median of all peptide ratios assigned to the protein. In addition a posterior error probability for each MS/MS spectrum below or equal to 0.1 was required. In case the identified peptides of two proteins were the same or the identified peptides of one protein included all peptides of another protein, these proteins were combined by MaxQuant and reported as one protein group. Before further statistical analysis, known contaminants and reverse hits were removed. For experiments 1 and 3, proteins with 'razor + unique peptides' >1 were selected. Further data analysis was performed using the Perseus software suite[32]. SILAC H:L ratios of two replicate experiments from the MaxQuant proteingroups.txt output table were imported into Perseus and statistical outliers were determined using standard one-sample two-sided $t$-testing (parameters: $p$-value < 0.05, S0: 0). Proteins displaying averaged H:L ratios >1.5 were selected for further analysis.

**Cell culture.** 129/Sv-Cast/EiJ[33] female ESCs were cultured as previously described[11]. In brief, ESCs were grown on feeder cells in DMEM (GIBCO), 15% standard foetal calf serum (FCS), 100 U ml$^{-1}$ penicillin/streptomycin, 0.1 mM NEAA, 0.1 mM 2-mercaptoethanol and 5000 U ml$^{-1}$ LIF.

For embryoid body differentiation, we plated $0.25 \times 10^6$ ESCs (day 0) in a 10-cm bacterial dish and let them differentiate in normal ESC medium with serum without LIF for 3, 6 and 15 days. At day 3 and day 6, embryoid bodies were disaggregated, and single cells were attached to slides by cytospin. At day 12, embryoid bodies were allowed to attach on gelatin-covered coverslips and grown for three more days at which point coverslips were processed for RNA FISH.

For monolayer differentiation, a confluent T25 was split in different concentrations and cells were differentiated in monolayer differentiation medium (IMDM + Glutamax (GIBCO), 15% FCS, 50 μg μl$^{-1}$ ascorbic acid, 0.1 mM NEAA, 100 U ml$^{-1}$ penicillin/streptomycin, 37.8 μl l$^{-1}$ monothioglycerol (97%)) in wells of 6-well dishes for different days with gelatin-coated round coverslips.

For blastocyst outgrowths, E3.5 blastocysts were removed of their zona pellucida with Acidic Tyrode's Solution (Sigma) and put in culture on gelatin-coated coverslips with normal ESC medium. During the next days, blastocysts attached and were grown for 4–5 additional days before proceeding to *Xist* RNA FISH and subsequent genotyping.

**Generation of KO ESCs using the CRISPR/Cas9 system.** The 20 nucleotide single guide RNA (sgRNA) sequences were designed using the online CRISPR Design Tool (http://crispr.mit.edu/, Zhang Feng Lab), and cloned into CRISPR/Cas9 nuclease plasmids (PX459, Addgene plasmid #48139) for *Rnf12* or CRISPR/Cas9 nickase plasmids (pX462, Addgene plasmid #62987) for *Rex1*.

To delete the entire *Rnf12* open reading frame, ESCs were electroporated with a pair of sgRNAs. Twenty-four hours after electroporation, the cells were selected in Puromycin for 32 h and cultured in normal ES medium for seven additional days, when clones were picked and expanded independently. Clones were karyotyped

and characterized for the presence of two X chromosomes and for the *Rnf12* deletion by restriction digestion and genomic sequencing and protein immunoblotting, respectively. To generate *Rnf12*$^{-/-}$*Rex*$^{+/-}$ and *Rnf12*$^{-/-}$*Rex1*$^{-/-}$ DKO ESCs, *Rnf12*$^{-/-}$ ESCs were electroporated with two pairs of sgRNAs targeting *Rex1*. Identical procedures as with the *Rnf12*$^{-/-}$ ESC generation were followed.

The following sgRNAs sequences were used in this study:
5′-GGAACAAGTACTCTAAACTA-3′ for *Rnf12*-target 1; 5′-AAAGCGCTGTACAAAAAGTT-3′ for *Rnf12*-target 2; 5′-CCGTGTAACATACACCATCC-3′ for *Rex1*-target 1; 5′-TCCACTCTGGTATTCTGGAC-3′ for *Rex1*-target 2; 5′-AAACCTTTCTCGCCAGGTTC-3′ for *Rex1*-target 3; 5′-CACTTCCTCCAAGCTTTCGA-3′ for *Rex1*-target 4.

**Generation of *Rex1* and *Rnf12* deficient mice.** All animal experiments were performed according to the legislation of the Erasmus MC Rotterdam Animal Experimental Commission.

The *Rex1* KO mice were generated from *Rex1*$^{+(129)/-(cas)}$ heterozygous KO ESC made by the bacterial artificial chromosome (BAC) targeting method. The targeting vector was designed to replace part of exon 4, the only coding exon of *Rex1*, with a neomycin cassette flanked by loxP sites. The two short chromosomal arms (975 bp and 782 bp) were PCR amplified using genomic DNA of a Cast/EiJ BAC (CH26-227-N6) as a template, cloned into pCR-BluntII-TOPO (Invitrogen) and linearized with NheI to introduce the kanamycin/neomycin cassette. Homologous recombination in bacteria was confirmed by PCR. The CH26-227-N6ΔRex1 BAC was linearized with PI-SceI digestion and electroporated into WT female ES cell line F1 2-1 (129/Sv-Cast/EiJ). ESC clones were isolated by selection with G418. Resistant clones were expanded for screening by PCR analysis. Correct targeting of the *Rex1* allele in ESC clones by homologous recombination was confirmed with primers amplifying a restriction fragment length polymorphism (RFLP) present in the 129/Sv allele, followed by XmnI digestion. The presence of two polymorphic X chromosomes was confirmed by amplification of a sequence located in the *Atrx* gene containing a RFLP (Pf1M1, present in the 129/Sv allele). Properly targeted clones had normal ESC morphology and karyotype. *Rex1*$^{+(129)/-(cas)}$ ESCs were injected into C57BL/6 host blastocysts and transferred into pseudopregnant C57Bl/6 females to generate chimeric founders. Female chimeric founders were crossed with C57BL/6 male mice to assess germline transmission. F1 heterozygous offspring were then backcrossed into 129/Sv or Cast/EiJ backgrounds for at least six generations. Primers used to construct the *Rex1* KO mice are listed in Supplementary Table 1. The *Rex1*-null ESC line was derived from *Rex1*$^{-/-}$ E3.5 blastocysts obtained from intercrosses of *Rex1*$^{-/-}$ 129 males and *Rex1*$^{-/-}$ cas females, following a standard protocol for ESC derivation.

To generate *Rnf12* KO mice, *Rnf12*$^{-(129)/-(cas)}$ ESCs[17] were used for blastocyst injections. Chimeric mice were crossed with C57BL/6 males. F1 *Rnf12*$^{-(cas)/y}$ males were crossed with female C57BL/6 mice. N1 *Rnf12*$^{-(b6)/-(cas)}$ mice were then crossed with male C57BL/6 mice from which *Rnf12*$^{-(cas)/y}$ males with a mostly cas chromosome were obtained (0–120 Mb). This male was subsequently used for backcrossing into Cast/EiJ and 129/Sv and experiments. Primers used for genotyping *Rnf12* mice are listed in Supplementary Table 1. All our ESC lines are mycoplasma free, confirmed by regular checks. No randomization of animals was used. The investigators were not blinded to group allocation of mice during the experiments.

To generate *Rex1*$^{-/-}$:*Hprt*$^{+/GFP}$ blastocysts, we crossed *Hprt*$^{+/GFP}$ mice[23] into our *Rex1* KO background. *Rex1*$^{-/-}$ *Hprt*$^{+/GFP}$ blastocysts were generated by crossing *Rex1*$^{-/-}$ females with *Rex1*$^{-/-}$:*Hprt*$^{GFP/y}$ males.

**ESCs IF staining.** ESCs were fixed for 10 min with 4% paraformaldehyde (PFA) at room temperature (RT) permeabilized for 10 min with 0.4% Triton X-100-PBS and blocked for 30 min with 10% goat serum in PBST (PBS with 0.05% Tween 20). Incubation with primary antibodies in blocking buffer was performed overnight at 4 °C. After washing with PBST, cells were incubated in blocking buffer with the secondary antibodies for 1 h at RT. Slides were then washed in PBST and mounted with ProLong® Gold Antifade Mountant with DAPI (Thermo Fisher Scientific). Images were acquired using a fluorescence microscope (Axioplan2; Carl Zeiss) or a confocal Zeiss LSM700 microscope (Carl Zeiss, Jena) with Zen image acquisition software. Images were processed with Fiji and CC Photoshop software (Adobe). The following primary antibodies were used: goat anti-REX1 (Santa Cruz, sc-50670, 1:50), mouse anti-OCT4 (Santa Cruz, sc-5279, 1:100), rabbit anti-RNF12 (a generous gift from Dr. Ingolf Bach, 1:100), rabbit anti-NANOG (Calbiochem, SC1000, 1:100) and rabbit anti-H3K27me3 (Diagenode, C15310069, 1:500). The following Alexa Fluor secondary antibodies were used: donkey anti-mouse 488 (Thermo Fisher Scientific, A-21202, 1:500), donkey anti-rabbit 488 (Thermo Fisher Scientific, A-21206, 1:500), donkey anti-rabbit 546 (Thermo Fisher Scientific, A10040, 1:500) and donkey anti-goat 647 (Abcam, ab150131, 1:500).

**Quantitative RT-PCR.** Total RNA was extracted from ESCs using TRI reagent (Sigma-Aldrich, T9424). In total, 1.5 μg of RNA from each sample was reverse transcribed into cDNA with random hexamer primers (Invitrogen) and Superscript III reverse transcriptase (Invitrogen) according to the manufacturer's instructions. Quantitative RT-PCR was performed using LightCycler® 480 SYBR Green I Master

(Roche) in a CFX384 real-time PCR detection system (Bio-Rad). *Actin* was used as a normalization control. All qPCR data represent the mean ± s.d. of three independent biological replicates, each performed in triplicate. The primer sequences used are listed in the Supplementary Table 3.

**Protein extraction and western blot**. To obtain nuclear extracts, cells were harvested in 1 ml ice-cold PBS plus complete protease inhibitor (Roche, 04693132001) and 15 µM MG132 (Sigma, C2211). Cell pellets were incubated with 400 µl buffer A (10 mM Hepes, 1.5 mM MgCl2, 10 mM KCl, 0.5 mM DTT, protease inhibitor, and 15 µM MG132) for 10 min on ice, vortexed for 30 s and centrifuged (2000 rpm/5 min/4 °C). Next, nuclei were lysed by adding 2× the pellet volumes of buffer C (20 mM Hepes, 25% glycerol, 420 mM NaCl, 1.5 mM MgCl2, 0.2 mM EDTA, 0.5 mM DTT, protease inhibitor and 15 µM MG132) for 20 min on ice, centrifuged (14,000 rpm/2 min/4 °C) and the supernatant was used as nuclear extract. Protein concentrations were determined using NanoDrop (Thermo Scientific). WB was performed using homemade SDS-PAGE gels and nitrocellulose membranes (Merck, GE10600002). Specific proteins were detected using goat anti-REX1 (Santa Cruz, sc-50670, 1:1000), rabbit anti-RNF12 (a generous gift from Dr. Ingolf Bach, 1:3000), mouse anti-RNF12 (Abnova, H00051132-B01P, 1:1000) and β-ACTIN (Santa Cruz, sc-1616, 1:1000) or β-Actin-Peroxidase (Sigma, A3854, 1:20,000) were used as a loading control. β-Actin-Peroxidase was detected using ECL Western blotting Detection Reagents (GE Healthcare) in an Amersham Imager 600 (GE Healthcare) detection system. Detection of all the other proteins was performed using Odyssey CLx imaging system with Image Studio 5.2 software (LI-COR Biosciences) with the corresponding secondary antibodies: IRDye 800CW donkey anti-rabbit, P/N 925-32213; IRDye 800CW donkey anti-goat, P/N 925-32214 and IRDye 680RD donkey anti-mouse, P/N 925-68072 (all from LI-COR Biosciences, 1:10,000). Uncropped images of the western blots in this paper are found in Supplementary Fig. 10.

**RNA FISH**. In brief, ESCs or differentiating ESCs were fixed for 10 min at RT with 4% PFA in PBS. Cells were subsequently washed three times with 70% EtOH for 3 min and permeabilized with 0.2% pepsin at 37 °C and re-fixed with 4% PFA RT for 5 min. Cells when then washed with PBS at RT and dehydrated with subsequent washes of 3 min with EtOH 70, 90 and 100%. Coverslips were then put on slides carrying the *Xist* hybridization probe. The probe was made as follows: 2 µg of a 5.5-kb BglII cDNA fragment covering exons 3–7 of mouse *Xist* was DIG-labelled (DIG Nick-Translation Kit, Roche) following the manufacturer's instructions. About 1 µl of the probe was diluted in hybridization mix (50% formamide, 2× SSC, 50 mM phosphate buffer (pH 7.0), 10% dextran sulphate) and 100 ng µl$^{-1}$ mouse Cot-1 DNA (Thermo Fisher Scientific). The probe was then denatured for 5 min at 95 °C and pre-hybridized for 45 min at 37 °C. The probe was then put on a slide on top of which the dry coverslip with cells was put on. Slides were then incubated overnight at 37 °C in a humid chamber filled with 50% formamide in 2× SSC buffer. The following day, slides were washed two times with 50% formamide—2× SSC at 42 °C and twice with RT TST (0.1 M Tris, 0.15 M NaCl, 0.05% Tween 20). Coverslips were then blocked with TSBSA (0.1 M Tris, 0.15 M NaCl, 2 mg ml$^{-1}$ BSA (Jena Bioscience, BU-102)) for 30 min at RT. The *Xist* probe detection was then performed by subsequent steps of three antibody incubations (Sigma-Aldrich, 11093274910, 1:500, and two FITC-labelled antibodies, Thermo Fisher Scientific 31627, 65-6111, 1:250, all in blocking buffer), 30 min at RT, each in the humid chamber filled with TST. Cells were washed three times for 5 min at RT with TST between each incubation and after the last incubation. Coverslips were then dehydrated as before and mounted with ProLong Gold Antifade with DAPI (Molecular Probes). Images were acquired using a confocal Zeiss LSM700 microscope (Carl Zeiss, Jena) with Zen image acquisition software and processed with Fiji and CC Photoshop software (Adobe).

**Blastocysts IF staining**. To collect blastocysts, 5- to 8-week-old female mice were superovulated by intraperitoneal (i.p.) injection of 5 IU of pregnant mare serum gonadotropin (Folligon; Intervet) followed by an i.p. injection of 5 IU of human chorionic gonadotropin (Chorulon; Intervet) 48 h later. These superovulated female mice were mated with selected males. Embryos at the blastocyst stage were harvested by flushing the uterus at 3.5 days post-coitum in M2 medium. When E4.5 blastocysts were needed, E3.5 blastocysts were cultured in SAGE-1 medium (Origio) under liquid paraffin (Origio) in 5% CO$_2$ at 37 °C for 24 h. For blastocyst IF staining, the zona pellucida was removed by incubation in Acidic Tyrode's Solution (Sigma) at RT for a few seconds. Afterwards, embryos were washed in M2 medium and fixed in 4% PFA in PBS for 20 min at RT. Subsequently, embryos were rinsed in PBS containing 0.01% v/v Tween-20 (P1379, Sigma) (PBS-T), permeabilized in PBS containing 0.2% Triton X-100 (23,472-9, Sigma) for 15 min on ice and blocked in blocking buffer (PBS-T, 2% w/v bovine serum albumin (BSA fraction V), 5% v/v normal goat serum) for 4 h at RT and incubated with appropriate primary antibodies diluted in blocking buffer at 4 °C overnight. The following antibodies were used in this study: goat anti-REX1 (Santa Cruz, sc-50670, 1:100), goat anti-KLF4 (R&D, AF3158, 1:400), goat anti-OCT4 (Santa Cruz, sc-8628, 1:400), mouse anti-RNF12 (Abnova, H00051132-B01P, 1:50), mouse anti-CDX2 mouse (BioGenex, MU392A-UC, 1:400), rabbit anti-NANOG (Calbiochem,

SC1000, 1:100) and rabbit anti-H3K27me3 (Diagenode, C15310069, 1:500). *Rex1*$^{-/-}$:*Hprt*$^{+/GFP}$ blastocysts were immunostained with mouse anti-GFP antibody (Roche, 11814460001, 1:2000). The following day, primary antibodies were rinsed three times in PBS-T and incubated with the appropriate secondary antibodies for 1 h at RT. The following Alexa Fluor secondary antibodies diluted in blocking buffer were used: donkey anti-mouse 647 (A-31571), donkey anti-goat 555 (A-21432), donkey anti-rabbit 488 (A-21206) (all from Thermo Fisher Scientific, 1:500) and donkey anti-goat 647 (Abcam, ab150131, 1:500). Embryos were then washed three times in blocking buffer, incubated briefly in increasing concentrations of Vectashield with DAPI (Vector laboratories) before mounting on poly-lysine slides in small drops of concentrated Vectashield with DAPI.

Confocal images were collected using a Zeiss LSM700 microscope (Carl Zeiss, Jena) and processed with Fiji and Adobe CC Photoshop software (Adobe). Cell counts were performed using Image J (Fiji) software.

When necessary, blastocysts were genotyped by PCR after IF/FISH and confocal microscopy. Briefly, embryos were individually recovered, washed in PBS and lysed in 10 µl of lysis buffer (AM1722, Cells-to-cDNA™ II Kit, Thermo Fisher Scientific) for 15 min at 75 °C. About 1 µl of the lysis solution was directly used in a 25-µl PCR reaction. Primer pairs used for the genotyping are listed in Supplementary Table 1.

**PGCs IF staining**. After embryo isolation from the uteri, regions containing the developing germ cells were dissected from E9.5 and E11.5 embryos. E9.5 embryo hinduts and E11.5 embryo trunks were fixed in ice cold 4% PFA for 3 h, followed by consecutive washes in PBS. Tissues were then processed for paraffin embedding using standard histology procedures and 5 µm paraffin sections were dissected with a Cryostat HM 560. Heat-mediated (900 W in a microwave for 20 min) epitope retrieval in citrate buffer pH 6.0 was performed on paraffin sections. After cooling down, sections were blocked with blocking solution (2% BSA, 5% donkey serum in PBS) for 30 min at RT, followed by primary antibody incubation at 4 °C overnight. The next day, slides were washed in PBS and incubated with secondary antibodies for 1 h at RT. Slides were then washed in PBS and mounted with ProLong® Gold Antifade Mountant with DAPI (Thermo Fisher Scientific). Confocal imaging was performed on a Zeiss LSM700 microscope (Carl Zeiss, Jena). The following primary antibodies were used: goat anti-OCT4 (Santa Cruz, sc-8628, 1:200) and rabbit anti-H3K27me3 (Diagenode, C15310069, 1:250). The following Alexa Fluor secondary antibodies were used: donkey anti-goat 555 1:400 and donkey anti-rabbit 488 1:250 (both from Thermo Fisher Scientific).

**RT-PCR analysis of mice tissues and embryos**. To assess XCI skewing, hybrid female mice (129/Sv-Cast/EiJ) were sacrificed by cervical dislocation. Parts of organs were collected, snap-frozen and triturated using micro-pestles in 1 ml of Trizol reagent (Invitrogen). RNA was purified following manufacturer's instructions; 1 µg RNA was DNase-treated (Invitrogen, #18068015) and reverse-transcribed with SuperScript II (Invitrogen, #18080051), using random hexamers. Allele-specific *Xist* expression was analysed by RT-PCR amplifying a length polymorphism using primers *Xist* LP. To determine the allele-specific X-linked gene expression of *Mecp2* and *G6pdx*, primers amplifying a DdeI RFLP in *Mecp2*, and primers amplifying a ScrFI RFLP in *G6pdx* were used. PCR products were digested with the indicated restriction enzymes and analysed on a 2% agarose gel stained with ethidium bromide. Allele-specific expression was determined by measuring relative band intensities using a Typhoon image scanner (GE healthcare) and ImageQuant TL software.

E11.5 embryos were obtained by natural matings and dissected from decidua in PBS. The endoderm (VYSE, imprinted XCI) and mesoderm (VYSM, random XCI) layers of the visceral yolk sac (VYS) were separated using the enzymatic method described in ref. [34]. Briefly, yolk sacs were dissected from E11.5 embryos and incubated at 4 °C for 1.5 h with gentle shaking in a mixture of 0.5% trypsin (Sigma, 59427 C) and 2.5% pancreatin (Sigma, P7545) in Ca$^{2+}$- and Mg$^{2+}$-free HANS balanced salt solution (Sigma, H6648), with Protector RNase Inhibitor (Sigma, 3335399001). After incubation, the yolk sacs were washed in PBS and the layers separated using fine forceps under a dissecting microscope. The VYSE layer was collected for RNA isolation using ReliaPrep RNA Cell Miniprep System (Promega). Similar procedures to analyse *Xist*, *Mecp2* and *G6pdx* as described above for mouse tissues were followed.

For *Tsix* expression analysis of single hybrid E3.5 blastocysts obtained from the crossing of a WT cas male with a *Rnf12*$^{+/-}$ 129 female, blastocysts were lysed in 10 µL of lysis buffer (AM1722, Cells-to-cDNA™ II Kit, Thermo Fisher Scientific). Strand- and allele-specific reverse transcription of *Tsix* and *Actin* was performed according to manufacturer's instructions, using 2 µM of the *Tsix*_A and *Actin* reverse primers. RT-PCR amplification of *Tsix* was then performed with primer pair *Tsix*_A followed by an additional round of nested PCR using *Tsix*_B primer pair. Amplified cDNAs were run on agarose gels and purified through the NucleoSpin Gel and PCR Clean-up columns (Macherey-Nagel). To distinguish between 129 and cas alleles, the purified DNA fragments were digested with restriction enzyme (MnlI RFLP), and electrophoresed in 3% agarose gels (Bio-rad, #1613107). RT-PCR amplification of *Actin* was performed with *Actin* primers. Primers used are listed in Supplementary Table 2.

## Data availability

The mass spectrometry proteomics data have been deposited to the Proteo-meXchange Consortium via the PRIDE[35] partner repository with the dataset identifier PXD010944. All other relevant data supporting the key findings of this study are available within the article and its Supplementary Information files or from the corresponding author upon reasonable request.

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

## Acknowledgements

We thank Federica Federici, Agnese Loda, Cheryl Maduro, Sarra Merzouk, Maud Borensztein, Mary Wallingford, Siyeon Rhee and Lies-Anne Severijnen for experimental advice. We also thank Helen Kock for excellent mouse care. This work was supported by grants from the Netherlands Organisation for Scientific Research (NWO-VENI) and the EUR grant from the Erasmus University of Rotterdam to C.G.

## Author contributions

C.G. and J.G. conceived the study. C.G., H.M.-B. and J.G. designed the experiments. C.G. and H.M.-B. performed most experiments, assisted by E.R. for the generation of the Rex1$^{+/-}$ ESCs and C.D. in the Rnf12$^{-/-}$ blastocysts IF staining. H.M.-B. and A.M. performed the primordial germ cell IF staining. C.G., T.S.B. and J.G. generated the Rnf12$^{-/-}$ ESCs used in the generation of the Rnf12$^{-/-}$ mice. C.G. and J.D. performed the mass spectrometry experiments. J.G., C.G. and H.M.-B. wrote the manuscript.

## Additional information

**Competing interests:** The authors declare no competing interests.

