## [Peer Review File · Nature Communications]

Reviewer #1 (Remarks to the Author):

Gontan et al. studied the role of Rex1 in Rnf12-mediated regulation of Xist. Based on the finding that Rex1 is a factor, the amount of which is solely increased in Rnf12^{-/-} ESCs, they suggested that Rex1 is a prime target of Rnf12 in ESCs. Then they showed that Rex1 deletion in Rnf12^{-/-} ESCs rescued the defect in random XCI, suggesting a crucial role of Rnf12-mediated degradation of Rex1 in the initiation of random XCI. They further generated Rex1-deficient mice and showed Rex1 was dispensable for not only X reactivation in ICM cells and PGCs but also imprinted and random XCI in vivo. When Rex1 deletion was combined with Rnf12 deletion, however, the lethal phenotype upon maternal transmission of Rnf12 deletion in Rnf12^{-/-} and Rnf12^{-/+} embryos were rescued, suggesting the importance of Rnf12-mediated degradation of Rex1 in the proper regulation of imprinted XCI.

The data provided basically suggest that Rnf12-mediated degradation of Rex1 plays a crucial role in imprinted as well as random XCI. The study brings about another piece of in vivo evidence that Rnf12 targets Rex1 for degradation to initiate not only random but also imprinted XCI. Although this study contains some interesting and important findings, I provide some suggestions below to further strengthen the paper.

Major points

1. They examined the effect of loss of Rex1 in Xi reactivation in E11.5 PGCs as well as in ICM of E3.5 and E4.5 embryos by monitoring the timing of H3K27me3 disappearance. However, I wonder if the idea that the loss of H3K27me3 in ICM cells and PGCs can be referred to as Xi reactivation has been previously shown to be appropriate or widely accepted. If it has been the case, then the authors should cite the relevant references. If not, they should avoid to definitely mention Rex1 is dispensable for Xi reactivation, or otherwise, should perform RNA-FISH to show expression of X-linked genes at two sites in the nucleus of ICM cells or PGCs. In addition, although comparison of H3K27me3 localization in ICM cells between E3.5 and E4.5 allows them to suggest no difference in the timing of Xi reaction, they would not be able to evaluate the effect of functional loss of Rex1 on Xi reactivation in PGCs by looking at just one stage (E11.5). Figure 3d only tells that the accumulation of H3K27me3 on Xi has been lost by E11.5 in both wildtype and Rex1^{-/-} PGCs. It is totally unclear if there is a difference between wildtype and Rex1^{-/-} PGCs in the timing when H3K27me3 starts or completes to be lost. The authors should compare localization of H3K27me3, preferably in combination with RNA-FISH for X-linked genes, in PGCs at different stages. I would also like to point out that the quality of immunostaining for H3K27me3 in PGCs should be improved.

2. For allele-specific expression analysis, how did the authors prepare the extraembryonic yolk sac? Did they separate the visceral endoderm from the yolk sac mesoderm? If they did not isolate the extraembryonic component of the yolk sac, they could not yet conclude that imprinted XCI was not affected.

3. Based on the observation that Rnf12⁻/Y males were underrepresented in the male pups born to Rnf12^{+/-} females crossed with wildtype males, they concluded Rnf12⁻/Y display partial embryonic lethality. It is not clear, however, how the sex and genotype of the offspring were identified. If the number of respective genotypes was based on the number of pups at weaning, the loss of Rnf12⁻/Y males could be attributable to not only embryonic but also postnatal lethality. The authors should clarify this.

4. In Discussion, the authors describe a scenario that Rnf12^{-/+} cells in the preimplantation embryos carrying the maternal deletion of Rnf12 initiate imprinted inactivation of their paternal wildtype X by upregulating Xist on it. This would lead to functional loss of Rnf12 as the paternal Rnf12 is effectively downregulated, resulting in accumulation of Rex1 protein, which they propose represses Xist initially upregulated on the paternal X. The authors expect that this ends up with derepression of the initially inactivated paternal X and lack of imprinted XCI after all. A series of previous work from the Gribnau lab suggest that dose of Rnf12 is critical for upregulation of Xist and one copy of Rnf12 is insufficient for stochastically upregulating Xist in random XCI in differentiating ESCs. If this is also the case in the cells of preimplantation embryos, wouldn't Rnf12^{-/+} cells upregulate paternal Xist anyway? Furthermore, I am also wondering if Rex1 protein accumulates in those cells lacking maternal Rnf12 in the preimplantation Rnf12^{-/+} embryos as were seen in the trophectoderm of Rnf12^{-/-} and Rnf12⁻/Y blastocysts.

5. The authors previously suggested that Rex1 was a potential activator of Tsix and Rnf12-mediated degradation of Rex1 abolished Tsix transcription, enabling Xist on the same chromosome to be expressed in cis to initiate XCI in ESCs. I am wondering if maternal transmission of Rnf12 deletion causes ectopic activation of paternal Tsix in Rnf12^{-/+} as well as Rnf12^{-/-} preimplantation embryos. If this is the case, one can ascribe the failure of imprinted XCI upon maternal transmission of Rnf12 deletion to ectopic expression of Tsix on the paternal X. The mice generated in this study provides a good opportunity to address this issue.

Minor points

1. Fig. 1d in line102 should read Fig. 1e.

2. Fig. 1e-g in line 105 should read Fig. 1d, f, and g.

3. In Figure 3a, 4b, and 5b, how did the authors confirm the genotype of the embryos? Did they extract DNA from each embryo after the observation by fluorescence microscopy for PCR?

4. Images of RNA-FISH and immunofluorescence are too small in general.

Reviewer #2 (Remarks to the Author):

The manuscript from Gribnau and colleagues addresses the mechanism by which RNF12, an E3-ubiquitin-ligase, functions in imprinted X chromosome inactivation. While RNF12 has been shown to be required for rXCI in mESCs, it is dispensable for rXCI in vivo. However, maternally-transmitted Rnf12-deletion causes iXCI defects and female embryonic lethality, indicating a specific role of Rnf12 in iXCI. This manuscript is a step forward in showing that REX1, a protein target of RNF12, is directly downstream of Rnf12 in the regulatory pathway for iXCI. By showing that iXCI defects in the Rnf12-deletion mutant embryo is associated with REX1 stabilization and that double-KO mice of Rnf12^{-/-}; Rex1^{-/-} are rescued, being viable and fertile irrespective of gender, the authors conclude that REX1 is downstream of RNF12 responsible for iXCI. In addition, the authors showed similar rescuing effects of REX1 removal in Rnf12^{-/-} mESCs, which exhibit the normal rXCI. Their model summary therefore includes mechanisms of Rnf12-Rex1 regulation for both iXCI and rXCI. Overall, this paper contains a large amount of work and data, and provides new insight into the regulatory pathways controlling Xist and XCI, which represent an important area of research that is still poorly understood. The knockout mutant mice and the analyses from this work would be a valuable resource for researchers in the field. This paper can make an important contribution to our understanding of regulatory factors in controlling the mouse XCI, if the authors clarify some specific points.

Major points:

1. To show that REX1 removal rescues Rnf12-deletion mouse viability, the most straightforward genetic cross should include a breeding of Rnf12^{-/-}; Rex1^{-/-} female with Rex1^{+/-} heterozygous male, in which case Rnf12^{-/+}; Rex1^{-/-} daughters will be viable due to Rex1^{-/-} rescue and their Rnf12^{-/+}; Rex1^{+/-} sisters will be nonviable due to a maternally-inherited Rnf12-deletion that is not rescued.
2. As the authors explained in the introduction, "iXCI takes place during preimplantation development in the embryo and in the extra-embryonic tissues, where the paternal X chromosome is always inactivated." However, Fig. 3f shows bi-allelic expression of X-linked genes in the yolk sac, for both WT and Rex^{-/-} samples. Similar observations are presented in Fig. 5d. Although the skewing

pattern in the yolk sac is obviously distinct from the expression pattern in the embryo, for a normal situation, shouldn't iXCI in yolk sac reflect an exclusively paternal-X silencing? A clarification would be helpful.

3. Fig. 2d shows that Xist level is increased in Rex1^{+/-} mESCs. Does it suggest that the "dose-sensitive" repressive role of Rex1 on Xist could be effective as a heterozygous Rex^{+/-} possibly leading to an ectopic expression of Xist? An Xist RNA FISH on these cells would be informative.

4. Fig. 3a: the image resolution is low. H3K27me3 staining in the extraembryonic cell looks confusing in the E3.5 Rex1^{-/-} sample. For E4.5 Rex1^{-/-} embryo, there seems to be two foci of H3K27me3 in a cell (e.g. cells surrounding the ICM in the zoom-in panel for Rex1^{-/-} show H3K27me3 pattern different from the WT example). It would be supportive to have Xist RNA FISH on these embryos, which should help clarify the XCI status.

5. Figs. 4b & 5b: What are Xist expression patterns in Rnf12^{-/-} embryos and in Rnf12^{-/-};Rex1^{-/-} embryo? Xist RNA FISH in these embryos should provide supportive and more direct evidence for the mutants' effect on Xist and iXCI.

6. Fig. 6: The "trans" repressive effect of RNF12 on REX1 for iXCI is confusing. With a trans effect, a paternal copy of RNF12 should be sufficient for REX1 degradation and iXCI in the Rnf12^{-/+} embryo. In the figure legend, Line 564: "This is followed by increased RNF12 expression in the developing epiblast, resulting in increased turnover of REX1, concomitant with its transcriptional downregulation, facilitating rXCI. In Rnf12^{-/+} and Rnf12^{-/-} embryos, REX1 levels are stabilised in the extra-embryonic tissues, preventing Xist upregulation and iXCI." The authors should incorporate these in the illustration – the model figure is too simplified and not very helpful for understanding.

7. In the discussion, the authors should address the possible involvement of other factors, which are identified in the same region of Xist. Specifically for imprinted XCI, transgenic studies have been reported for factors in the X-inactivation center, which is directly responsible for the mouse iXCI.

8. It would be helpful if the authors could clarify the usage and analysis of different KO ESC lines, i.e. generated by CRISPR or HR-targeting, in regard to the consistence or possible difference in phenotypes

Minor points:

1. Higher-resolution images should be used in the figures panel: e.g. Fig. 3a&c, Supplementary Fig. 3a, Supplementary Fig. 5a.
2. Line 105: (Fig. 2d,e) does not support “Rnf12 is required for rXCI in vivo” as the figured was referred.
3. Supplementary Figure 4: This figure title does not match the content.

Reviewer #3 (Remarks to the Author):

In this work, Gontan et. al. extend their previous findings (Ref 11) on the regulation of Xist and X inactivation by RNF12 and REX1 from mES cells to a mouse model. The present study shows that the removal of Rex1 is sufficient to rescue embryonic lethality of Rnf12 knockout, suggesting that REX1 is required for the maintenance of iXCI during development.

I recommend publication after the following points are addressed in the manuscript.

1. In figure 1, the figure labels of 1d and 1e are swapped, leading to inconsistency between the description within the main text, figure legend and the figures; The legend for Fig 1. is also wrong.
2. Line 102-3 Listing of figures does not match the text in Fig 1. (e.g. western blot is 1e, not 1d, also for qPCR and FISH)
3. For SILAC results, it is common practice in the proteomics field to present all of the data points on the scatter-plot Fig 1b, rather than just a single quadrant and schematized figure. Doing so will provide readers a clearer understanding of the quality of the SILAC pulldown experiment and readers can visualize if the target protein is differentially expressed. This is especially so when no statistic test was performed as was the case in this manuscript.

Dear reviewers,

We would like to thank you for your constructive comments and questions. Besides the requested experiments and clarifications we have also included novel data related to the Rnf12 and Rex1 knockout ES cell lines described and studied in this manuscript. In our present manuscript we have studied Rnf12^{neo/-puro} ESCs previously generated by homologous recombination (Barakat 2011) with alleles that leave the 5' region of Rnf12 intact, resulting in a 333aa peptide which lacks the catalytic Ring finger and interaction domain. In addition, for our study, we have now generated by CRISPR/Cas9 targeting new Rnf12^{CR-/CR-} ESCs that lack the complete open reading frame of Rnf12 (Fig. 1b,c and Supplementary Fig. 2). A recent paper published by the Bach laboratory (Wang Cell Reports 2017) suggests that the presence of the 333aa peptide in our Rnf12^{neo/-puro} ESCs results in nuclear accumulation of REX1, possibly explaining discrepancies with respect to the XCI phenotype observed between both laboratories. To clarify this issue, we performed Western blotting (WB), IF and XCI studies comparing wild type (WT) ESCs with Rnf12^{neo/-puro} and Rnf12^{CR-/CR-} ESCs in the same genetic background. Our WB analysis confirms the presence of the truncated 333aa RNF12 peptide in Rnf12^{neo/-puro} ESCs and the complete absence of RNF12 protein in Rnf12^{CR-/CR-} ESCs and that REX1 is accumulated at equally high levels in Rnf12^{neo/-puro} and Rnf12^{CR-/CR-} ESCs as compared to the WT ESCs (Supplementary Figure 2a). In contrast to what is reported by Wang et al., immunocytochemistry analysis indicates nuclear accumulation of REX1 both in Rnf12^{neo/-puro} and Rnf12^{CR-/CR-} knockout ESC lines (Fig. 1d and Supplementary Figure 3b). Finally, examination of XCI dynamics indicates no difference between both Rnf12 knockout ESC lines, showing loss of XCI upon monolayer as well as embryoid body differentiation assays (Supplementary Fig. 3c-g). These findings indicate that the Rnf12^{neo/-puro} alleles behave as the full Rnf12 knockout alleles, which we think is important to show and discuss in the revised manuscript. We have included these results in Fig1 and Supplementary Fig2 and 3, and describe our results in the accompanying text and discussion.

For this revision, we have generated a new homozygous null Rex1^{-/-} ESC line (129/Sv:Cast/Eij) from Rex1^{-/-} blastocysts. We have performed WB and XCI studies comparing WT ESCs with Rex1^{+/-} and Rex1^{-/-} ESCs. (Fig. 2c,e and f, and Supplementary Fig.5). We observe an increase in the percentage of cells with two Xist clouds during differentiation in both Rex1^{+/-} and Rex1^{-/-} ESCs. This result highlights the importance of Rex1 as a repressor of XCI.

Below we address these comments in a point by point rebuttal.

Reviewer 1

Major points

1. They examined the effect of loss of Rex1 in Xi reactivation in E11.5 PGCs as well as in ICM of E3.5 and E4.5 embryos by monitoring the timing of H3K27me3 disappearance. However, I wonder if the idea that the loss of H3K27me3 in ICM cells and PGCs can be referred to as Xi reactivation has been previously shown to be appropriate or widely accepted. If it has been the case, then the authors should cite the relevant references. If not, they should avoid to definitely mention Rex 1 is dispensable for Xi reactivation, or otherwise, should perform RNA-FISH to show expression of X-linked genes at two sites in the nucleus of ICM cells or PGCs. In addition, although comparison of

H3K27me3 localization in ICM cells between E3.5 and E4.5 allows them to suggest no difference in the timing of Xi reaction, they would not be able to evaluate the effect of functional loss of Rex1 on Xi reactivation in PGCs by looking at just one stage (E11.5). Figure 3d only tells that the accumulation of H3K27me3 on Xi has been lost by E11.5 in both wildtype and Rex1^{-/-} PGCs. It is totally unclear if there is a difference between wildtype and Rex1^{-/-} PGCs in the timing when H3K27me3 starts or completes to be lost. The authors should compare localization of H3K27me3, preferably in combination with RNA-FISH for X-linked genes, in PGCs at different stages. I would also like to point out that the quality of immunostaining for H3K27me3 in PGCs should be improved.

Susana M. Chuva de Sousa Lopes and Anne McLaren (PLoS Genetics 2008) and Mariana de Napoles et al (Plos One 2007), thoroughly studied the relationship between loss of H3K27me3, Xist and X-reactivation in PGCs. Their work indicates that loss of H3K27me3 happens concomitantly with loss of Xist, preceding reactivation which happens shortly afterwards. On a gene by gene basis X reactivation is heterogeneous with some genes being reactivated shortly after loss of H3K27me3/Xist, others later during PGC development (Sugimoto and Abe Plos Genetics 2007), indicating that a global marker such as H3K27me3 will be more useful than focussing on single genes.

Nevertheless, we have attempted to follow X reactivation of a single transgene by crossing in X-linked GFP and tdTomato alleles located in the Hprt locus (Wu Neuron 2014). Reactivation of this reporter was not detected in migrating PGCs both in a WT and Rex1^{-/-} background indicating that reactivation of this reporter happens later after they reach the female gonads. Unfortunately, expression of the transgene in germ cells was down-regulated in the ovary and therefore reactivation (presence of yellow cells-GFP⁺/Tomato⁺) could not be assessed in this setting (see rebuttal Fig. 1).

Figure 1. E9.5 and E14.5 Hprt^{CAG-Tomato/CAG-GFP} transgenic embryos were isolated and PGCs were identified with anti OCT4 and TRA98 antibodies, respectively. No reactivation of Hprt-reporters (presence of yellow cells-GFP⁺/Tomato⁺) is detected in migrating PGCs at E9.5 (top panels) and 11.5 (not shown). In E14.5 ovaries the Hprt-reporter is silenced, and therefore reactivation cannot be studied.

With respect to the timing of reactivation we agree with this reviewer that we should have looked at different time points to conclude on X reactivation dynamics. We therefore also examined PGCs in

E9.5 WT, *Rex1*^{-/-} and *Rnf12*^{-/-}*Rex1*^{-/-} embryos determining the percentage of cells containing H3K27me3 coated X chromosomes. This study indicates that X chromosome reactivation is not delayed in the PGCs, and even shows a trend towards enhanced reactivation in the *Rex1*^{-/-}. We have included these new panels in Fig. 3c,d, Supplementary Fig. 6d,e and supplementary Fig. 9b of our revised manuscript, and have modified the text accordingly. With respect to the immunostainings quality we apologize, due to journal restrictions the resolution was low in the submitted figures. The revised manuscript contains higher resolution pictures.

Similar studies have been performed by the Lee and other laboratories focussing on the ICM, also showing a tight relationship between loss of *Xist* and H3K27me3 followed by Xi reactivation (Payer Mol Cell 2014). Like in PGCs, different genes are reactivated in the ICM at different stages during preimplantation development (Borensztein Nature Communications 2017), for this reason, instead of focusing on a single gene, we decided to make use of our X-linked GFP allele located in the *Hprt* locus (Wu Neuron 2014) to follow Xi reactivation in the epiblast lineage of the ICM, a similar approach has been used in a recent study to visualize Xi reactivation dynamics (Kobayashi Development 2016). E4.5 blastocysts obtained from crosses of *Rex1*^{-/-} females with *Rex1*^{-/-}:*Hprt*^{GFP/y} males, were analysed by GFP and H3K27me3 IF (Supplementary Fig.6 b,c). Our results show that GFP detection in the epiblast, signaling the reactivation of the paternal Xi, is tightly linked to the loss of H3K27me3 and no difference was observed between *Rex1*^{-/-} and WT female blastocysts.

2. For allele-specific expression analysis, how did the authors prepare the extraembryonic yolk sac? Did they separate the visceral endoderm from the yolk sac mesoderm? If they did not isolate the extraembryonic component of the yolk sac, they could not yet conclude that imprinted XCI was not affected.

We agree with the reviewer. In the revised manuscript, we have repeated our experiments and performed the trypsin-pancreatin method (Nagy 2003) to enzymatically separate the mesoderm (VYSM, embryonic layer with random XCI) and endoderm (VYSE, extra-embryonic layer with imprinted XCI) layers of the visceral yolk sac (VYS). RT-PCR using the placental marker *Afp* and vascular endothelial marker *Flk-1* confirmed that the separation had been successful (data not shown). We again find that *Rex1*^{-/-} and *Rnf12*^{-/-}:*Rex1*^{-/-} have identical imprinted XCI compared to WT. Fig. 3e,f; 5c,d and Supplementary Fig. 7c and 9d have been modified accordingly. See material and methods section for detailed protocol.

3. Based on the observation that *Rnf12*^{-/Y} males were underrepresented in the male pups born to *Rnf12*^{+/-} females crossed with wildtype males, they concluded *Rnf12*^{-/Y} display partial embryonic lethality. It is not clear, however, how the sex and genotype of the offspring were identified. If the number of respective genotypes was based on the number of pups at weaning, the loss of *Rnf12*^{-/Y} males could be attributable to not only embryonic but also postnatal lethality. The authors should clarify this.

Sexing and genotyping of pups was performed at day 5 after birth. We therefore agree with this reviewer that loss of *Rnf12*^{-/y} pups could have happened perinatally or early postnatally. We have adapted the manuscript accordingly.

4. In Discussion, the authors describe a scenario that *Rnf12*^{-/+} cells in the preimplantation embryos carrying the maternal deletion of *Rnf12* initiate imprinted inactivation of their paternal wildtype X by upregulating *Xist* on it. This would lead to functional loss of *Rnf12* as the paternal *Rnf12* is effectively downregulated, resulting in accumulation of *Rex1* protein, which they propose represses *Xist* initially upregulated on the paternal X. The authors expect that this ends up with derepression of the initially inactivated paternal X and lack of imprinted XCI after all. A series of previous work from the Gribnau lab suggest that dose of *Rnf12* is critical for upregulation of *Xist* and one copy of *Rnf12* is insufficient for stochastically upregulating *Xist* in random XCI in differentiating ESCs. If this is also the case in the cells of preimplantation embryos, wouldn't *Rnf12*^{-/+} cells upregulate paternal *Xist* anyway?

Our previous work has indicated that the robustness of XCI is affected in *Rnf12*^{+/-} cells. Nevertheless, *in vitro* and *in vivo* many cells are still capable of initiating XCI, indicating that more XCI activators are involved in XCI initiation (Jonkers Cell 2009, Barakat PLoS Genetics 2010, Shin Nature 2010). We also showed that at least one intact copy of *Rnf12* is required for XCI to be initiated *in vitro* (Barakat PLoS Genetics 2010, Barakat Mol Cell 2014) and that at least one copy of *Rnf12* is required for sufficient break down of REX1. Our present study confirms this model for both in iXCI in the embryo and rXCI in ESCs. In iXCI, loss of XCI in *Rnf12*^{-/-} mice can be rescued by removal of *Rex1* resulting in live *Rnf12*^{-/-}:*Rex1*^{-/-} animals, indicating that accumulation of REX1 is the main factor in iXCI mediated lethality. In *Rnf12*^{+/-} blastocysts, iXCI is mostly abrogated as can be observed by the absence of H3K27me3 domains in trophoblasts cells. However, in some cells, *Xist* manages to escape REX1 repression and inactivates the X chromosome. This is not sufficient to sustain proper embryonic development, and *Rnf12*^{+/-} eventually die. With respect to rXCI, we also find a rescue of the XCI phenotype observed in differentiating *Rnf12*^{-/-} ESCs upon removal of *Rex1*. However, for this revised manuscript we also established *Rex1*^{-/-} ES cell lines and determined and found that the percentage of cells with two clouds was significantly increased in differentiating *Rex1*^{-/-} ESCs. This finding, included as panels e and f in Figure 2 of the revised manuscript, and described in the accompanying text and discussion, supports a role for the RNF12:REX1 axis in negative feedback, preventing XCI of the second X chromosome. We also modified the model presented in Figure 6 to provide a better overview of our findings.

Furthermore, I am also wondering if *Rex1* protein accumulates in those cells lacking maternal *Rnf12* in the preimplantation *Rnf12*^{-/+} embryos as were seen in the trophoctoderm of *Rnf12*^{-/-} and *Rnf12*^{-/Y} blastocysts.

As shown in Supplementary Figure 8b, *Rnf12*^{-/+} preimplantation embryos do not show increased levels of REX1 as *Rnf12*^{-/-} and *Rnf12*^{-/Y} embryos. This might be due to the fact that the paternal X chromosome cannot be properly inactivated otherwise the cells would become *Rnf12*^{-/-} which cannot upregulate *Xist*. Indeed, most of the cells in the trophoblast of *Rnf12*^{-/+} blastocysts do not show H3K27me3 domains, which suggest these embryos have problems with inactivating their paternal Xi. Essentially, most of the trophoblast cells still keep the paternal X active and express RNF12 which will keep the levels of REX1 too low to enable the detection of REX1 upregulation by immunostaining. In the few cells that start Xi on the paternal X, as soon as RNF12 gets inactivated REX1 levels will be transiently upregulated leading to repression of *Xist* and ineffective imprinted XCI.

This negative feedback loop that prevents imprinted XCI to take place in the *Rnf12*^{-/+} embryos is facilitated by the proximity along the X chromosome of *Rnf12* and *Xist* genes, *Rnf12* is one of the first genes to get compliantly inactivated on the Xp, as early as at the 8 cells stage only maternal *Rnf12* expression is detected (Borensztein Nature Communications 2017). In addition, according to the results of our SILAC experiments, RNF12 and REX1 have a high *turn-over* rate, which will make the regulation of imprinted XCI via the RNF12-REX1 axis a very dynamic and fast process.

5. The authors previously suggested that *Rex1* was a potential activator of *Tsix* and *Rnf12*-mediated degradation of *Rex1* abolished *Tsix* transcription, enabling *Xist* on the same chromosome to be expressed in cis to initiate XCI in ESCs. I am wondering if maternal transmission of *Rnf12* deletion causes ectopic activation of paternal *Tsix* in *Rnf12*^{-/+} as well as *Rnf12*^{-/-} preimplantation embryos. If this is the case, one can ascribe the failure of imprinted XCI upon maternal transmission of *Rnf12* deletion to ectopic expression of *Tsix* on the paternal X. The mice generated in this study provides a good opportunity to address this issue.

We have addressed this question in the new Supplementary Fig. 8c. of our revised manuscript, and have modified the text accordingly. E3.5 hybrid embryos (129:cas) were generated by crossing 129 *Rnf12*^{+/-} females mice with cas WT male mice. Allele specific *Tsix* expression was studied by RT-PCR making use of a *Tsix* RFLP (MnlI)(Lee Cell 2000) that produced a 129-specific band (198 bp) and a cas band (151 bp) (see materials and methods section for more details). During the revision period, we encountered breeding problems with our Cast/EiJ *Rnf12* KO mice, which is why we could only confirm that there is no ectopic expression of the paternal *Tsix* (cas allele) in *Rnf12*^{-(129)/+(cas)} female embryos. Therefore, the failure of iXCI is due to direct repression of *Xist* and not to the upregulation of *Tsix* from the paternal allele. During rXCI in the differentiation of *Rex1*^{+/-} and *Rex1*^{-/-} ESC lines *Xist* expression is up-regulated and *Tsix* expression is down-regulated in a REX1 dose-dependent mode (Supplementary Fig. 5).

Minor points

1. Fig. 1d in line102 should read Fig. 1e.

Adapted accordingly.

2. Fig. 1e-g in line 105 should read Fig. 1d, f, and g.

Adapted accordingly.

3. In Figure 3a, 4b, and 5b, how did the authors confirm the genotype of the embryos? Did they extract DNA from each embryo after the observation by fluorescence microscopy for PCR?

Yes, when necessary, the genotyping of the blastocysts was performed by PCR after immunostaining/FISH and confocal microscopy. Briefly, embryos were individually recovered, washed in PBS and lysed in 10 µl of lysis buffer (AM1722, Cells-to-cDNA™ II Kit, Thermo Fisher Scientific) for 15min at 75°C. 1 µl of the lysis solution was directly used in a 25 µl PCR reaction. Primer

pairs used for the genotyping are listed in Supplementary Table 2. A description of the genotyping has been implemented in the Materials & Methods section of the revised manuscript.

4. Images of RNA-FISH and immunofluorescence are too small in general.

As requested by this reviewer we have enlarged our Figure panels when possible.

Reviewer #2 (Remarks to the Author):

1. To show that REX1 removal rescues Rnf12-deletion mouse viability, the most straightforward genetic cross should include a breeding of Rnf12^{-/-}; Rex1^{-/-} female with Rex1^{+/-} heterozygous male, in which case Rnf12^{-/+}; Rex1^{-/-} daughters will be viable due to Rex1^{-/-} rescue and their Rnf12^{-/+}; Rex1^{-/+} sisters will be nonviable due to a maternally-inherited Rnf12-deletion that is not rescued.

These new crossings have been included as Supplementary Figure 9a. We have performed two different types of informative crossings: females Rnf12^{+/-}:Rex1^{-/-} X males Rex1^{+/-} and females Rnf12^{-/-}:Rex1^{-/-} X males Rex1^{+/-}. In neither crossings were Rnf12^{+/-}:Rex1^{-/-} females born, while their Rnf12^{-/-}:Rex1^{-/-} sisters were.

2. As the authors explained in the introduction, “iXCI takes place during preimplantation development in the embryo and in the extra-embryonic tissues, where the paternal X chromosome is always inactivated.” However, Fig. 3f shows bi-allelic expression of X-linked genes in the yolk sac, for both WT and Rex^{-/-} samples. Similar observations are presented in Fig. 5d. Although the skewing pattern in the yolk sac is obviously distinct from the expression pattern in the embryo, for a normal situation, shouldn't iXCI in yolk sac reflect an exclusively paternal-X silencing? A clarification would be helpful.

We agree with the reviewer. Our previous visceral yolk sac (VYS) isolation protocol resulted in contamination of the extraembryonic tissue with embryonic material. In the revised manuscript, we have repeated our experiments and performed the trypsin-pancreatin method (Nagy 2003) to enzymatically separate the mesoderm (VYSM, embryonic layer with random XCI) and endoderm (VYSE, extra-embryonic layer with imprinted XCI) layers of the VYS. RT-PCR using the placental marker Afp and vascular endothelial marker Flk-1 confirmed that the separation had been successful (data not shown). We again find that Rex1^{-/-} and Rnf12^{+/-}:Rex1^{-/-} mice have identical imprinted XCI compared to WT. Fig. 3e,f; 5c,d and Supplementary Fig. 7c and 9d have been modified accordingly. In the newly isolated VYSE there is a total skewing pattern where Xist expression is exclusively paternal and G6pdx and Mecp2 are maternally expressed.

3. Fig. 2d shows that Xist level is increased in Rex1^{+/-} mESCs. Does it suggest that the “dose-sensitive” repressive role of Rex1 on Xist could be effective as a heterozygous Rex^{+/-} possibly leading to an ectopic expression of Xist? An Xist RNA FISH on these cells would be informative.

As suggested by this reviewer, we investigated *Xist* expression in WT, *Rex1*^{+/-} and the newly generated *Rex1*^{-/-} ESCs by qPCR and *Xist* RNA FISH. Our qPCR results show a dose-dependent increase in *Xist* expression during differentiation of *Rex1*^{+/-} and *Rex1*^{-/-} ESCs cells as compared to WT (Supplementary Fig. 5). In line with this result, we observed by *Xist* RNA FISH that, *Rex1*^{+/-} and *Rex1*^{-/-} ESCs cells display significantly more *Xist* clouds at day 3 and 6 of differentiation. More importantly, we also find ectopic *Xist* expression with a significant increase in the number of cells displaying two clouds (Fig. 2e,f). This indicates that *Rex1* acts as an important break on the system involved in the feedback mechanism preventing XCI of too many X chromosomes. This finding is described in the results section and the discussion of the revised manuscript.

4. Fig. 3a: the image resolution is low. H3K27me3 staining in the extraembryonic cell looks confusing in the E3.5 *Rex1*^{-/-} sample. For E4.5 *Rex1*^{-/-} embryo, there seems to be two foci of H3K27me3 in a cell (e.g. cells surrounding the ICM in the zoom-in panel for *Rex1*^{-/-} show H3K27me3 pattern different from the WT example). It would be supportive to have *Xist* RNA FISH on these embryos, which should help clarify the XCI status.

We apologize for the limited quality of our Figures, which was related to the size limits of this journal. We have improved the quality of the pictures, and included panels with a larger magnification. The blastocyst confocal images in these figures are representative maximum intensity projections of a Z-stack. The counting of the H3K27me3 domains has been performed by ImageJ on the separate optical sections out of the confocal stack. After careful analysis of the separate optical sections we can conclude that the different clouds belong to separate cell nuclei. To show iXCI in a more clear way, we have also performed *Xist* RNA-FISH on WT, *Rex1*^{-/-} and *Rnf12*^{-/-}/*Rex1*^{-/-} blastocyst outgrowths as performed by others (Shin 2014 and Wang 2017), which show no differences with WT blastocysts outgrowths in terms of iXCI in the trophoblast cells (Supplementary Fig. 7a,b).

5. Figs. 4b & 5b: What are *Xist* expression patterns in *Rnf12*^{-/-} embryos and in *Rnf12*^{-/-};*Rex1*^{-/-} embryo? *Xist* RNA FISH in these embryos should provide supportive and more direct evidence for the mutants' effect on *Xist* and iXCI.

We have performed *Xist* RNA-FISH on *Rnf12*^{-/-}/*Rex1*^{-/-} blastocyst outgrowths and show that there are no differences with WT embryos (Supplementary Fig. 7d,e). The *Xist* expression pattern in *Rnf12*^{-/-} embryos has been already studied by others (Bach Nature 2010).

6. Fig. 6: The “trans” repressive effect of RNF12 on REX1 for iXCI is confusing. With a trans effect, a paternal copy of RNF12 should be sufficient for REX1 degradation and iXCI in the *Rnf12*^{-/+} embryo. In the figure legend, Line 564: “This is followed by increased RNF12 expression in the developing epiblast, resulting in increased turnover of REX1, concomitant with its transcriptional downregulation, facilitating rXCI. In *Rnf12*^{-/+} and *Rnf12*^{-/-} embryos, REX1 levels are stabilised in the extra-embryonic tissues, preventing *Xist* upregulation and iXCI.” The authors should incorporate these in the illustration – the model figure is too simplified and not very helpful for understanding.

We apologize for the confusion and have adapted Figure 6 accordingly.

7. In the discussion, the authors should address the possible involvement of other factors, which are identified in the same region of Xist. Specifically for imprinted XCI, transgenic studies have been reported for factors in the X-inactivation center, which is directly responsible for the mouse iXCI.

We agree with this reviewer that, besides Rnf12 and Rex1, other factors are involved in the regulation of iXCI. Among these factors are the repressive imprint of the maternal Xist copy (as transgene and other studies have indicated), but may also include Yy1 mediated activation of the paternal copy of Xist, analogous to the role of Yy1 in regulation of rXCI. We have included this possible mechanism in our revised discussion section.

8. It would be helpful if the authors could clarify the usage and analysis of different KO ESC lines, i.e. generated by CRISPR or HR-targeting, in regard to the consistence or possible difference in phenotypes

As suggested by this reviewer we have included an additional Supplementary Figure panel (Supplementary Figure 2a) describing all Rnf12 and Rex1 alleles that have been described in this manuscript.

Minor points:

1. Higher-resolution images should be used in the figures panel: e.g. Fig. 3a&c, Supplementary Fig. 3a, Supplementary Fig. 5a.

We did.

2. Line 105: (Fig. 2d,e) does not support “Rnf12 is required for rXCI in vivo” as the figured was referred.

We apologize and have modified the text accordingly.

3. Supplementary Figure 4: This figure title does not match the content.

We have changed the title of this Figure.

Reviewer #3 (Remarks to the Author):

In this work, Gontan et. al. extend their previous findings (Ref 11) on the regulation of Xist and X inactivation by RNF12 and REX1 from mES cells to a mouse model. The present study shows that the removal of Rex1 is sufficient to rescue embryonic lethality of Rnf12 knockout, suggesting that REX1 is required for the maintenance of iXCI during development.

I recommend publication after the following points are addressed in the manuscript.

1. In figure 1, the figure labels of 1d and 1e are swapped, leading to inconsistency between the description within the main text, figure legend and the figures; The legend for Fig 1. is also wrong.

We apologize for the confusion and have modified the legend accordingly.

2. Line 102-3 Listing of figures does not match the text in Fig 1. (e.g. western blot is 1e, not 1d, also for qPCR and FISH)

We apologize for the confusion and have modified the legend accordingly.

3. For SILAC results, it is common practice in the proteomics field to present all of the data points on the scatter-plot Fig 1b, rather than just a single quadrant and schematized figure. Doing so will provide readers a clearer understanding of the quality of the SILAC pulldown experiment and readers can visualize if the target protein is differentially expressed. This is especially so when no statistic test was performed as was the case in this manuscript.

As suggested by this reviewer, all identified proteins in our SILAC-based quantifications are shown in the new scatterplots (Figure 1a and Supplementary Figure 1d). Statistical analysis was now carried out using Perseus software (standard one-sample two-sided t-testing) and depicted as volcano plots in Supplementary Figure 1b and e. See material and methods section for detailed information.

Reviewer #1 (Remarks to the Author):

I found the manuscript has been improved very much and the authors have satisfactorily addressed and responded all the concerns I raised in the previous round of review. The concept that Rnf12 plays a crucial role in the initiation of random as well as imprinted XCI by targeting Rex1 for proteasomal degradation should be shared by researchers in broad fields.

Reviewer #2 (Remarks to the Author):

The authors have added substantial experiments and clarifications to the revised manuscript, which have addressed my previous concerns. The newly generated Rex1^{-/-} ES cell lines with upregulation of Xist and two Xist clouds are interesting and supportive of Rex1's role as a repressor of Xist in differentiating ESCs.

However, in the in vivo mouse model, the authors clearly demonstrated and emphasized that Rex1^{-/-} had no effect on either iXCI or rXCI in mouse embryos. I would recommend that the authors add a few sentences to help clarify the perceived inconsistency between consequences of Rex1 mutations in ESCs and embryos. This would benefit the model of Rnf12-Rex1.

In addition, since the adapted model (new Figure 6) has clarified the direct regulatory effect of Rnf12 on Xist through Rex1 for iXCI, which reflect the main findings of this paper through the rescue of Rnf12^{-/-} by Rex1^{-/-} in mice, I still believe that a comparison to Xist expression in Rnf12^{-/-} embryos is most relevant to the direct evidence of Rex^{-/-} rescuing Rnf12^{-/-} defects in both Xist regulation and iXCI. The authors addressed that "The Xist expression pattern in Rnf12^{-/-} embryos has been already studied by others (Bach Nature 2010)." In that case, the published finding should be described and referenced with the interpretation of rescuing effects by Rex1^{-/-}.

Reviewer #3 (Remarks to the Author):

The authors have addressed my concerns. I recommend publication.

Dear reviewers,

We would like to thank you for your constructive comments. We have addressed below both comments raised by Reviewer #2:

Reviewer #2 (Remarks to the Author):

The authors have added substantial experiments and clarifications to the revised manuscript, which have addressed my previous concerns. The newly generated Rex1^{-/-} ES cell lines with upregulation of Xist and two Xist clouds are interesting and supportive of Rex1's role as a repressor of Xist in differentiating ESCs.

However, in the in vivo mouse model, the authors clearly demonstrated and emphasized that Rex1^{-/-} had no effect on either iXCI or rXCI in mouse embryos. I would recommend that the authors add a few sentences to help clarify the perceived inconsistency between consequences of Rex1 mutations in ESCs and embryos. This would benefit the model of Rnf12-Rex1.

In our revised submission, we have implemented the Discussion section with additional text to clarify the perceived inconsistency.

In addition, since the adapted model (new Figure 6) has clarified the direct regulatory effect of Rnf12 on Xist through Rex1 for iXCI, which reflect the main findings of this paper through the rescue of Rnf12^{-/-} by Rex1^{-/-} in mice, I still believe that a comparison to Xist expression in Rnf12^{-/-} embryos is most relevant to the direct evidence of Rex^{-/-} rescuing Rnf12^{-/-} defects in both Xist regulation and iXCI. The authors addressed that "The Xist expression pattern in Rnf12^{-/-} embryos has been already studied by others (Bach Nature 2010)." In that case, the published finding should be described and referenced with the interpretation of rescuing effects by Rex1^{-/-}.

We have added this information in the Discussion section. Shin et al 2010 showed by RNA FISH that trophoblasts cells of Rnf12^{-/-} blastocysts do not have Xist clouds. We believe this is due to the stabilization of REX1 that we observe in our Rnf12^{-/-} blastocysts, as we have shown in our manuscript. The Shin et al 2010 results fit with our model: removal of RNF12, leads to stabilization of REX1 and concomitant repression of Xist in trophoblasts cells. We now explain this issue in more detail and refer to the manuscript of the Shin et al 2010 results in our Discussion.